# Generation of functional ciliated cholangiocytes from human pluripotent stem cells

Mina Ogawa [1], Jia-Xin Jiang[2,10], Sunny Xia[2,10], Donghe Yang[1,10], Avrilynn Ding[1], Onofrio Laselva[2], Marcela Hernandez[1], Changyi Cui[1], Yuichiro Higuchi[3], Hiroshi Suemizu [3], Craig Dorrell [4], Markus Grompe [4], Christine E. Bear[2,5,6] & Shinichiro Ogawa [1,7,8,9 ✉]

The derivation of mature functional cholangiocytes from human pluripotent stem cells (hPSCs) provides a model for studying the pathogenesis of cholangiopathies and for developing therapies to treat them. Current differentiation protocols are not efficient and give rise to cholangiocytes that are not fully mature, limiting their therapeutic applications. Here, we generate functional hPSC-derived cholangiocytes that display many characteristics of mature bile duct cells including high levels of cystic fibrosis transmembrane conductance regulator (CFTR) and the presence of primary cilia capable of sensing flow. With this level of maturation, these cholangiocytes are amenable for testing the efficacy of cystic fibrosis drugs and for studying the role of cilia in cholangiocyte development and function. Transplantation studies show that the mature cholangiocytes generate ductal structures in the liver of immunocompromised mice indicating that it may be possible to develop cell-based therapies to restore bile duct function in patients with biliary disease.

[1] McEwen Stem Cell Institute, University Health Network, Toronto, ON, Canada. [2] Programme in Molecular Medicine, Research Institute, Hospital for Sick Children, Toronto, ON, Canada. [3] Central Institute for Experimental Animals, Kawasaki, Kanagawa, Japan. [4] Oregon Stem Cell Center, Oregon Health and Science University, Portland, OR, USA. [5] Department of Physiology, University of Toronto, Toronto, ON, Canada. [6] Department of Biochemistry, University of Toronto, Toronto, ON, Canada. [7] Ajmera Transplant Centre, Toronto General Research Institute, University Health Network, Toronto, ON, Canada. [8] Department of Surgery, Shinshu University School of Medicine, Matsumoto, Nagano, Japan. [9] Department of Laboratory Medicine and Pathobiology, University of Toronto, Toronto, ON, Canada. [10] These authors contributed equally: Jia-Xin Jiang, Sunny Xia, Donghe Yang.
✉ email: Shinichiro.Ogawa@uhnresearch.ca

The liver is comprised of many different cell types that interact to carry out the broad spectrum of functions performed by the adult organ. One of these functions, the production of bile necessary for lipid digestion is orchestrated between the hepatocytes that secrete it and the biliary system that transports it from the liver to the intestines. The biliary system consists of a series of interconnected bile ducts that are formed from a specialized population of epithelial cells known as cholangiocytes[1]. In addition to providing the structure of the bile ducts, cholangiocytes play an active role in bile flow as they are responsible for regulating its viscosity and osmolality as it moves through the liver[2]. Cholangiocytes carry out this function through a complex sensing mechanism mediated by a series of receptors, ion channels and sensory signaling molecules found on a structure known as a primary cilia that extends from the apical membrane of the cell into the lumen of the bile duct[3]. The primary cilia on the cholangiocytes function as mechano, chemo and osmo-sensors and in response to changes in bile flow or composition trigger signaling pathways that stimulate fluid secretion or absorption from the cholangiocytes[4–6]. The manipulation of bile composition through this ciliated-mediated sensing mechanism is essential for maintaining liver homeostasis[7].

Diseases that impact cholangiocyte function are known as cholangiopathies and include monogenic disorders such as cystic fibrosis (CF) and Alagille syndrome as well as those with unknown etiology such as biliary atresia, primary biliary cirrhosis, and primary sclerosing cholangitis[8,9]. Although these are different diseases, they share in common the fact that they all disrupt bile transport, resulting in hepatocyte toxicity, compromised organ function and in extreme cases, organ failure. The only treatment for end stage disease is organ transplantation[10]. One of the most common cholangiopathies, cystic fibrosis (CF), is caused by a mutation in the cystic fibrosis transmembrane conductance regulator (CFTR) that encodes a chloride ion channel expressed by the cholangiocyte and responsible for the regulation of bile secretion. Mutations in CFTR result in the deposition of viscous, acidic bile that is toxic to the surrounding hepatocytes, leading to liver damage and organ failure[11,12]. As the life expectancy of CF patients increases with the availability of new treatments that improve lung function, cystic fibrosis liver disease (CFLD) is emerging as a major cause of the morbidity and mortality in these patients[11]. Small molecule drugs that rescue CFTR function have been identified and recently approved for clinical use by the FDA and other regulatory agencies[13]. These drugs have been shown to improve lung function of patients carrying the common F508del CFTR mutation, but patient–patient variability highlights an unmet need for patient-specific screening strategies for the identification of new CF drugs[14–16]. Additionally, as the drugs are largely evaluated on lung function their effect on other organs remains to be determined. Recent studies have focused on the use of CF patient organoids generated from primary intestinal, lung, and nasal epithelial cells as an approach to measure both patient and tissue-specific therapeutic responses of CFTR modulators[17–19]. However, limited access to primary cholangiocytes precludes efforts to establish comparable high throughput biliary organoid screens to identify drugs that target CFTR function in these cells.

To overcome the limitation of accessibility of primary cholangiocytes, a number of groups, including ours, have turned to human pluripotent stem cells (hPSCs) as a source of these cells for modeling cholangiopathies[20–25]. In our previous study, we showed that it was possible to specify the cholangiocyte lineage from a hPSC-derived hepatoblast population through staged activation of the Notch pathway, recapitulating the observation in vivo that Notch signaling is required for establishing the cholangiocyte fate[21]. When cultured in semi-solid media with HGF, EGF, and TGFβ, these cells formed cholangiocyte cysts that displayed characteristics of rudimentary biliary structures including the ability to respond to agonists and drugs that activate CFTR. Other studies have similarly reported on the development of staged protocols using either monolayer or 3D culture formats to differentiate cholangiocyte-like cells from hPSCs. A number of pathways, including IL-6[22], EGF[20,23,24], TGFβ and/Notch[25] were manipulated at different times in the cultures, yielding cells that expressed genes and proteins associated with the cholangiocyte lineage including SOX9, CK7, CK19, CFTR, and SLC4A2 (Anion Exchange protein 2; AE2), among others. While these studies demonstrate that it is possible to generate cholangiocyte-like cells from hPSCs, they all have limitations which include inefficient differentiation and low cell yield and incomplete maturation. Notably, none of the studies to date reported the efficient development of ciliated cholangiocytes. This is a major drawback as it indicates that the cells are not fully mature and precludes studies on the role of cilia in normal cholangiocyte function and in the pathophysiology of different cholangiopathies.

To address this shortcoming, we developed of monolayer-based differentiation strategy that yields high numbers of mature, ciliated cholangiocytes from different hPSC lines. Using several different screen strategies, we identify the retinoic acid, BMP, cAMP, and Rho kinase pathways as key regulators of cholangiocyte maturation. The cells generated with this approach express high levels of CFTR appropriate for drug screening and contain primary cilia, that in response to fluid flow, initiate intracellular $Ca^{2+}$ release and activate CFTR. Following transplantation into the spleen of TK-NOG immunocompromised recipients, the hPSC-derived cholangiocytes migrated to the liver where they form multiple ductal structures consisting of cells that display characteristics of mature cholangiocytes. With these advances, it is now possible to engineer biliary structures with appropriate staged ciliated cholangiocytes to model different cholangiopathies and to develop innovative drug- and cell-based therapies to treat them.

## Results

**RA signaling promotes development of CFTR$^+$ cholangiocytes.** As a first step to generate mature functional cholangiocytes, we investigated the role of different signaling pathways on the specification of the cholangiocyte fate from day 27 hepatoblasts generated with our previously published protocol. For these studies, the hepatoblasts were plated in the presence of HGF and EGF for 4 days either on OP9 jagged-1 mouse stromal cells (OP9j) to induce Notch signaling or on Matrigel without stromal support. Subsequently, agonists and antagonists of pathways known to promote early bile duct differentiation were added and the levels of CFTR, SOX9, alpha fetoprotein (AFP), and albumin (ALB) were measured following 6 days of culture. (Fig. 1a, b). Cells cultured on OP9j cells without any additional factors expressed higher levels of the early cholangiocyte marker SOX9 and lower levels of the hepatocyte markers ALB than those cultured on Matrigel alone, indicating that Notch signaling induced the initial stages of cholangiocyte development in the monolayer cultures. Analysis of the populations treated with the different pathway regulators revealed that only retinoic acid (RA) signaling induced significant levels of CFTR expression. Although expression was upregulated in cells cultured in both formats, the levels were dramatically higher (3- to 4-fold higher) in those co-cultured with OP9j. RA did not affect the levels of SOX9 but did further reduce the expression of AFP and ALB (Fig. 1b). Western blot analyses revealed that RA concentrations of 500 nM, 1 μM, and 2 μM all induced significant levels of CFTR protein in the

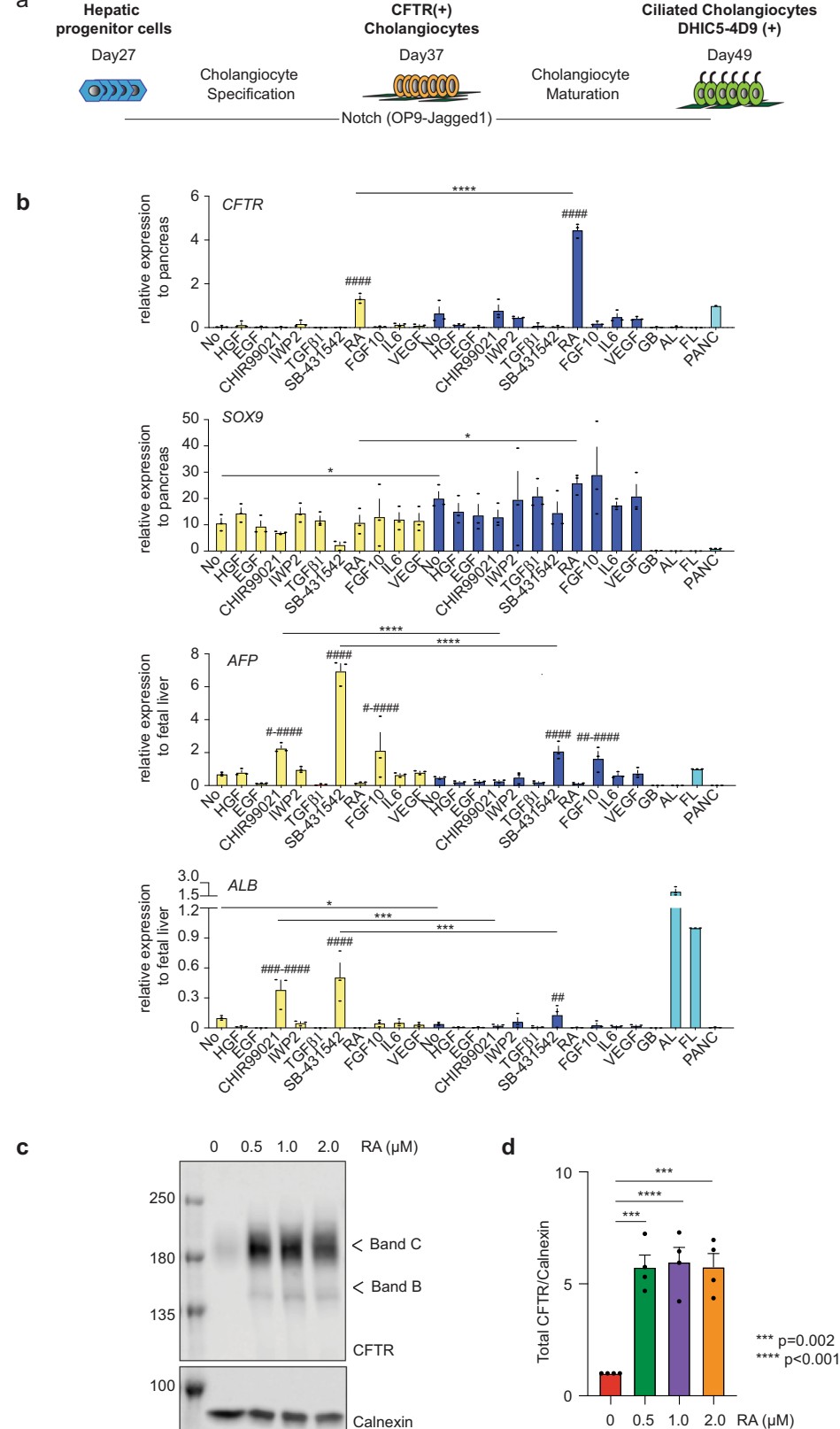

developing cholangiocytes (Fig. 1c, d). Collectively, these findings indicate that RA signaling plays a role in the specification of the cholangiocyte lineage as demonstrated by the upregulation of *CFTR* and in the inhibition of the hepatocyte program as shown by the downregulation of the hepatoblast and hepatocyte markers *AFP* and *ALB*.

**Generation of ciliated cholangiocytes from hPSCs.** Although RA promoted the development of CFTR[+] cholangiocytes in monolayer cultures, <20% of the cells were ciliated, indicating that most were not fully mature. To identify regulators that would promote further maturation of the CFTR[+] cholangiocytes, we developed a flow cytometry based screening approach using the

**Fig. 1 Establishment of differentiation protocol of hPSC-derived functional cholangiocytes in monolayer culture. a** Schematic of differentiation protocol.
**b** RTqPCR analysis of *CFTR*, *SOX9*, *AFP*, and *ALB* expression in the presence of factors for 6 days following HGF and EGF treatment for 4 days after the
passage on either Matrigel (yellow) or OP9-Jagged1 (blue) from the day 27 hepatoblast. GB gall bladder, AL adult liver, FL fetal liver, PANC pancreas.
#Represents statistical significance among factors in Matrigel or in OP9-Jagged1. #$p \leq 0.05$, ##$p \leq 0.01$, ###$p \leq 0.001$, ####$p \leq 0.0001$ one-way ANOVA.
* Represents statistical significance between factors in Matrigel and in OP9-Jagged1 *$p \leq 0.05$, ***$p \leq 0.001$, ****$p \leq 0.0001$ two-tailed Student's *t* test. Data
are represented as mean ± SEM. (*n* = 3). **c** Western blot analysis showing mature and immature glycosylated CFTR bands C and B in hESC-derived
cholangiocytes at day 37 after treated with different concentrations of RA. Numbers on the left side represent the molecular weight. Uncropped blots in
Source Data. **d** Quantification of CFTR proteins in hPSC-derived cholangiocytes at day 37 after treated with different concentrations of RA. two-tailed
Student's *t* test compared to RA 0μM. Data are represented as mean ± SEM. (*n* = 4).

antibody DHIC5-4D9 that stains adult human bile and pancreatic
ducts[21,26,27] but not the day 37 RA-induced hPSC-derived cho-
langiocytes (Fig. 2a). We analyzed agonists and antagonists of
pathways known to play a role in bile duct development and
maturation[28–34] for their ability to induce the generation of
DHIC5-4D9+ cells in the hPSC-derived cholangiocyte popula-
tion. As shown in Fig. 2b, Supplementary Fig. 1a, treatment with
either a Rho-kinase inhibitor (RI), Forskolin (FSK), cAMP, or the
BMP inhibitor Noggin (NOG) promoted the development of
DHIC5-4D9+ cells at day 43 of culture. The proportion of
positive cells in the RI-, FSK-, cAMP-, and NOG-treated groups
was: 30.9 ± 4.5%, 26.6 ± 4.6%, 24.3 ± 4.3%, and 18.0 ± 2.1%
(mean ± SEM) respectively (Fig. 2b). The combination of NOG,
FSK, and RI (NFR) was more effective than each factor alone and
generated populations that contained 68.0 ± 4.0% and 79.9 ± 3.7%
DHIC5-4D9+ cells at days 43 and 49 of culture respectively
(Fig. 2a and Supplementary Fig. 1b). The population of DHIC5-
4D9+ cells significantly decreased in the absence of RI, suggesting
that it is the main contributor to DHIC5-4D9+ cholangiocyte
differentiation (Fig. 2a). NFR did not induce the DHIC5-4D9+
population if RA was inhibited with BMS493, indicating that RA
signaling is required for the early stages of cholangiocyte
maturation and cannot be replaced by NOG, FSK, and RI
(Fig. 2c). As previous studies have shown that EGF signaling
plays a role in the maturation of cholangiocytes from hPSCs[20–25],
we included it in our screen. EGF did not promote the devel-
opment of DHIC5-4D9+ cells under the conditions of our cul-
tures. Moreover, when combined with NFR, it inhibited the
generation of DHIC5-4D9+ cells (Supplementary Fig. 1b).

Molecular analyses revealed changes in gene expression within
the RA/NFR-treated population indicative of bile duct develop-
ment and maturation. RA alone induced the upregulation of
expression of the ductal genes *SPP1*, *CK7*, and *SOX9*, of genes
that encode the hormone receptor *SCTR* and the ion channels
*SLC4A2*, *CFTR*, and of those that encode proteins involved in
cilia formation and function such as *TRPV4*, *PKD1*, *PKD2* in the
day 37 populations (Fig. 2d). Although RA signaling promoted
these changes, the levels of some of the genes, most notably those
associated with cilia development and function, were not
maintained over the following 12 days in the absence of NFR.
The addition of NFR at day 37 led to even higher levels of *CFTR*,
*SCTR*, and *SPP1* expression by day 49 than those induced by RA
alone (Fig. 2d). Additionally, NFR maintained the RA-induced
levels of cilia related genes in the day 49 populations. The RA/
NFR induced population expressed CFTR, CK7, SLC4A2,
SCTR (Secretin Receptor), SLC10A2 (NTCP2/Apical Sodium-
Dependent Bile duct Transporter/ASBT), CK19 proteins indicat-
ing that these cells represent functional cholangiocytes (Fig. 2e
and Supplementary Fig. 2a). The observation that the primary
cilia and CFTR were expressed on the apical side of the cells
suggests that they acquire polarity in vitro (Supplementary
Fig. 2b–d). Flow cytometry analyses confirmed the immunos-
taining and molecular analyses and showed that the majority of
the RA/NFR-treated cells expressed CK7 and EPCAM. Very few

ALB+ cells were detected in the population (Supplementary
Fig. 3a).

To demonstrate CFTR function, we measured apical chloride
conductance (ACC) using a membrane potential dye (FLIPR
assay)[18]. CFTR function following the NFR treatment (day 49)
was higher than that immediately following RA treatment (day
37) indicating that the cholangiocytes matured over this period of
time in the presence of NFR (Supplementary Fig. 3b).

To determine if the induction of the DHIC5-4D9+ population
in hESC (H9) derived cholangiocytes correlated with the
development of cells with primary cilia, we stained the RA/
NFR-treated cells with an antibody against acetylated α- tubulin
(Figs. 2e and 3a, and Supplementary Fig. 2a and Supplementary
Video 1). As shown in Fig. 3b, the majority of cells in the
population were ciliated. Quantification of different populations
revealed that RA treatment alone promoted cilia development in
~15% of cells at day 37, and 28% of cells at day 49 in the absence
of NFR. In contrast, over 85% (87.2 ± 2.5%, mean ± SEM) of the
RA/NFR-treated cells were ciliated at day 49 indicating that
DHIC5-4D9 staining correlated with cilia development as a
parameter for cholangiocyte maturation. The addition of EGF in
place of NFR did not increase the proportion of ciliated positive
cells over that observed in the population treated with RA alone.
Further analyses showed that the cilia on the cells expressed
ARL13B, a Ras superfamily GTPase known to be present on
primary cilia of mature cholangiocytes[35] (Fig. 3c and Supple-
mentary Fig. 2a). SEM showed the presence of both primary cilia
and microvilli, indicating that the ultrastructure of the hPSC-
derived cholangiocytes was similar to of primary adult
cholangiocytes[36] (Fig. 3d).

To formally demonstrate that DHIC5-4D9 staining is a marker
of maturation, we isolated the DHIC5-4D9+ and DHIC5-4D9-
fractions from a day 49 population (Supplementary Fig. 4) and
conducted qRT-PCR on them. As shown in Fig. 3e, the DHIC5-
4D9+ cells expressed higher levels of *CFTR*, *TRPV4*, *PKD1*, and
*PKD2* and lower levels of *ALB* and *AFP* than the DHIC5-4D9−
cells. These findings demonstrate that the DHIC5-4D9+ cells
represent a more mature stage of cholangiocyte development than
the DHIC5-4D9− cells and in doing so validate the use of the
DHIC5-4D9 antibody for monitoring cholangiocyte maturation.
In addition to promoting maturation, treatment with NFR
increased the yield of ciliated cholangiocytes on day 49 to an
average of 2.15 ± 0.12 per day 27 hepatoblast or 11.38 ± 1.73 per
input hPSC. In contrast, RA-induced populations without NFR
treatment generated on average 0.05 ± 0.03 ciliated cholangiocytes
per input hepatoblast on day 49 of culture (Supplementary
Fig. 5a). Taken together, the findings from this series of studies
demonstrate that it is possible to efficiently generate functional
ciliated cholangiocytes in a monolayer format with appropriate
staged manipulation of specific signaling pathways.

**Measuring CF drug responses with hPSC-derived cholangio-
cytes.** To determine if the cholangiocytes generated with the
monolayer protocol can be used to evaluate the effects of

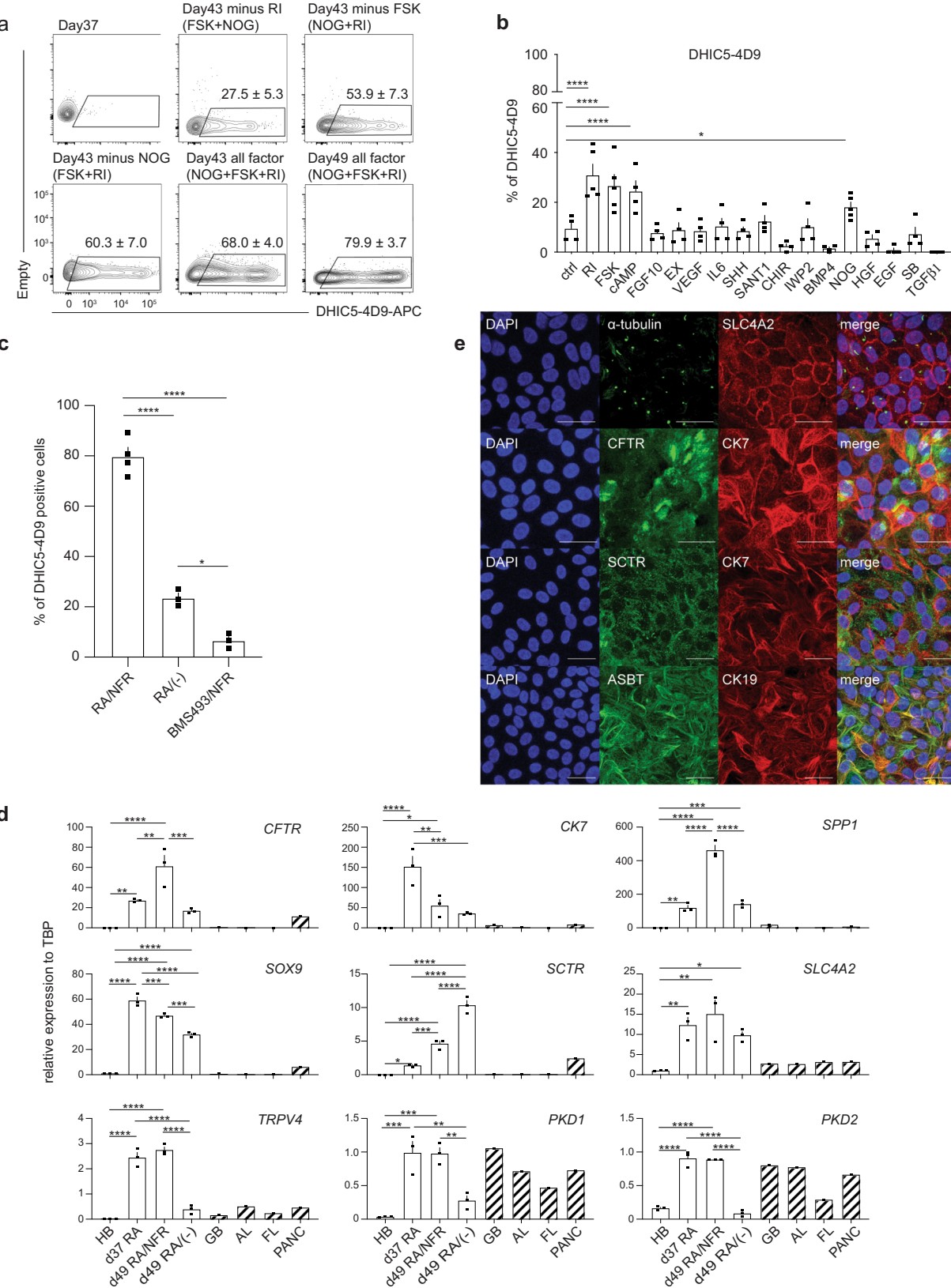

modulators on CFTR function, we measured the Z prime score on H9 hPSC-derived cells using the FLIPR assay in 96-well culture formats. This analysis revealed a score of 0.478 following RA treatment (Supplementary Fig. 5b), and 0.629 following NFR treatment (Fig. 4a). These high scores indicate that the FLIPR assay should be able to reliably predict responses to CFTR

modulators designed to rescue CFTR function in cholangiocytes generated from CF iPSCs.

To test this, we used the monolayer protocol to generate cholangiocytes from iPSCs derived from three different CF patients carrying the common F508del mutation (CF01, CF02, and GM4320[37]). For comparison, the CFTR activity of cholangiocytes

**Fig. 2 FSK, NOG, and RI promote the generation of DHIC5-4D9 positive cells. a** Flow cytometry analysis showing the proportion of DHIC5-4D9+ cells at day 37, day 43, and day 49. All factor represents addition of Noggin, Forskolin and Rho-Kinase inhibitor (NFR) following RA treatment. Minus represents withdrawal of each representative factor from NFR for 6 days. Data are represented as mean ± SEM ($n = 3$–5). **b** % Positivity of DHIC5-4D9+ cells after 6 days in the presence of factors. $^{*}p \leq 0.05$, $^{****}p \leq 0.0001$ two-tailed Student's $t$ test compared to control. Data are represented as mean ± SEM ($n = 5$ biologically independent samples from RI, FSK, NOG, $n = 4$ biologically independent samples per group from the rest). **c** Quantification of DHIC5-4D9 positive cells in day 49 hPSC-derived cholangiocytes after treatment with different indicated cytokine/small molecule combinations. $^{*}p \leq 0.05$, $^{****}p \leq 0.0001$ one-way ANOVA. Data are represented as mean ± SEM ($n = 4$ biologically independent samples from RA/NFR, $n = 3$ biologically independent samples per group from the rest). **d** RT-qPCR analysis of the expression of the indicated gene in each population at different stages of H9-derived cholangiocytes. HB hepatoblast, GB gall bladder, AL adult liver, FL fetal liver, PANC pancreas. $^{*}p \leq 0.05$, $^{**}p \leq 0.01$, $^{***}p \leq 0.001$, $^{****}p \leq 0.0001$ one-way ANOVA. Data are represented as mean ± SEM ($n = 3$). **e** Immunostaining analysis showing the proportion of acetylated α-tubulin and SLC4A2 (top), CFTR and CK7 (second row from the top), SCTR and CK7 (third from the top), ASBT and CK19 (bottom) cells in day 49 cholangiocyte. Scale bar represents 25 μm.

generated from a wild-type hES line, an iPSC line (WT01), and genome edited lines (CF01MC and CF02MC) were also evaluated. All iPSC lines had normal karyotypes and displayed characteristics of iPSCs including SSEA4, Tra-1-60, SOX2, NANOG, and OCT4 by protein expression (Supplementary Figs. 6–9). These patient lines differentiated efficiently and generated populations highly enriched for CXCR4+cKIT+EPCAM+HDE1+[38] endoderm by day 9 of differentiation (Supplementary Fig. 10a). The endoderm from each of the CF iPSC lines differentiated to give rise to hepatoblasts (Supplementary Fig. 10b) and subsequently to DHIC5-4D9+ cholangiocytes that expressed genes indicative of mature ciliated cells (Supplementary Fig. 11a, b). Staining for acetylated α-tubulin revealed that the RA/NFR induced populations from all of the lines contained a high frequency of ciliated cells (range 75–90%) indicating that the approach to promote maturation in a monolayer format is applicable to different hPSC lines (Supplementary Fig. 11c).

To validate the use of the in vitro generated cholangiocytes for drug evaluation, the patient iPSCs and control H9 hESCs were differentiated to mature cholangiocyte populations in microtiter wells, then treated with different CFTR modulators 24 h prior to analyses of apical chloride conductance (ACC) using the FLIPR assay that provides a measure of electrogenic chloride efflux. The modulators tested include small molecule combinations of approved drugs such as VX770, VX809 + VX770, VX661/VX445 + VX770 as well as emerging CFTR modulators developed by Abbvie: AC1 (X281602), AC2-1 (X281632), AC2-2 (X300549), or combinations of them with the potentiator drug AP2 (X300529)[39]. H9 hESC-derived cholangiocytes showed a high ACC response to FSK stimulation (Fig. 4b). Importantly, this response was inhibited by addition of the CFTR inhibitor CFTRinh-172, confirming that the assay is measuring CFTR channel activity. Western blot analysis confirmed robust expression of CFTR protein in the H9 hESC-derived cholangiocytes (Fig. 4c). The cholangiocytes from the patient iPSCs showed different responses to different combinations of modulators. Although iPSCs from two different patients (CF01, CF02) carried the same F508 del mutation, the cholangiocytes generated from them responded differently to a number of the small molecule combinations including VX809 + VX770 that has been approved for treatment. The triple combination of CFTR modulators AC1/AC2-1 + AP2, AC1/AC2-2 + AP2, or VX661/VX445 + VX770 promoted better CFTR activity than VX809 + VX770 rescuing the CFTR function to levels that were ~75% of those observed in the CFTR mutation corrected lines. To evaluate the contribution of clonal variation to the observed drug responses, we generated cholangiocytes from two different clones from each of the CF01 (a, b) and CF02 (a, b) iPS lines and tested them in FLIPR assay. We observed similar responses of the clone-derived cholangiocytes to drug combinations AC1/AC2-1 + AP2 (CF01, CF02) and VX661/VX445 + VX770 (CF02). Some differences were observed in response to VX809 + VX770 and AC1/AC2-2 + AP2 in CF02

(Fig. 4d–i). As expected from FLIPR response, generation of the complex glycosylated, mature form of the CFTR was enhanced by treatment with the corrector molecules, including VX809, VX661/VX445, AC1, and AC2-1 and AC2-2 (Supplementary Fig. 12). These results demonstrate that the cholangiocytes generated with the monolayer culture described here can be used to screen for drugs that modulate CFTR function in CF patient-derived cells.

**Generation of 3D cyst/organoids from monolayer cholangiocyte populations**. We previously described an approach for generating hPSC-derived 3D cholangiocyte cysts from hepatoblasts by culturing the cells in semi-solid media[21]. Although these cysts showed a Forskolin (FSK) induced CFTR-mediated swelling response, the format of the cultures did not permit easy access to the cysts for further analyses. The ability to generate organoids in liquid from the monolayer cultures would represent a technically simpler and more efficient method to produce these structures. Additionally, it would also provide two complimentary platforms for measuring CFTR responses from the same population: ACC using the FLIPR assay and FSK-induced cyst swelling. To generate organoids, we dissociated day 49 cholangiocyte monolayers with collagenase to form aggregates and cultured them on a rotating platform in 10 cm non-adherent dishes. Following 6 days of culture, we observed the development of hollow cysts that were remarkably homogenous in size and shape (Fig. 5a). Immuno-fluorescent analyses showed that cysts were comprised of CK19 or CK7+ cells that expressed CFTR and primary cilia (Fig. 5b, c).

Molecular analyses showed that the cholangiocytes in the monolayers and cysts at day 55 expressed different levels of genes that encode channel and transporter proteins essential for bile duct function. The monolayer cells expressed higher levels of *CFTR*, and the calcium-activated chloride channel *TMEM16* than the cysts whereas expression of *AQP1* that encodes a water channel protein and *SLC5A1* that encodes a sodium/glucose transporter was much higher in the cysts (Supplementary Fig. 13). Expression levels of *SLC4A2*, the anion exchange protein 2 (*AE2*), and the sodium/potassium/chloride transporter *SLC12A2* were comparable in both populations.

When tested in the swelling assay, H9 hESC-derived cholangiocyte cysts showed an average $1.16 \pm 0.01$-fold increase in size in response to FSK within 60 min (Fig. 5d and Supplementary Video 2). This acute response was blocked by the addition of CFTR inhibitor, confirming that the swelling was associated with CFTR function. Secretin hormone also induced fluid secretion resulting in a $1.09 \pm 0.01$-fold increase in cysts size (Fig. 5d and Supplementary Video 3). Analyses of the GM4320- and CF01 mutant-derived cysts revealed a significant swelling response ($1.07 \pm 0.01$ and $1.09 \pm 0.01$-fold respectively) to the drug combination of modulators (AC1/AC2-2 + AP2), compared to the FSK treated control (Fig. 5e, f). The response of the cholangiocytes from the mutation corrected CF01 cells

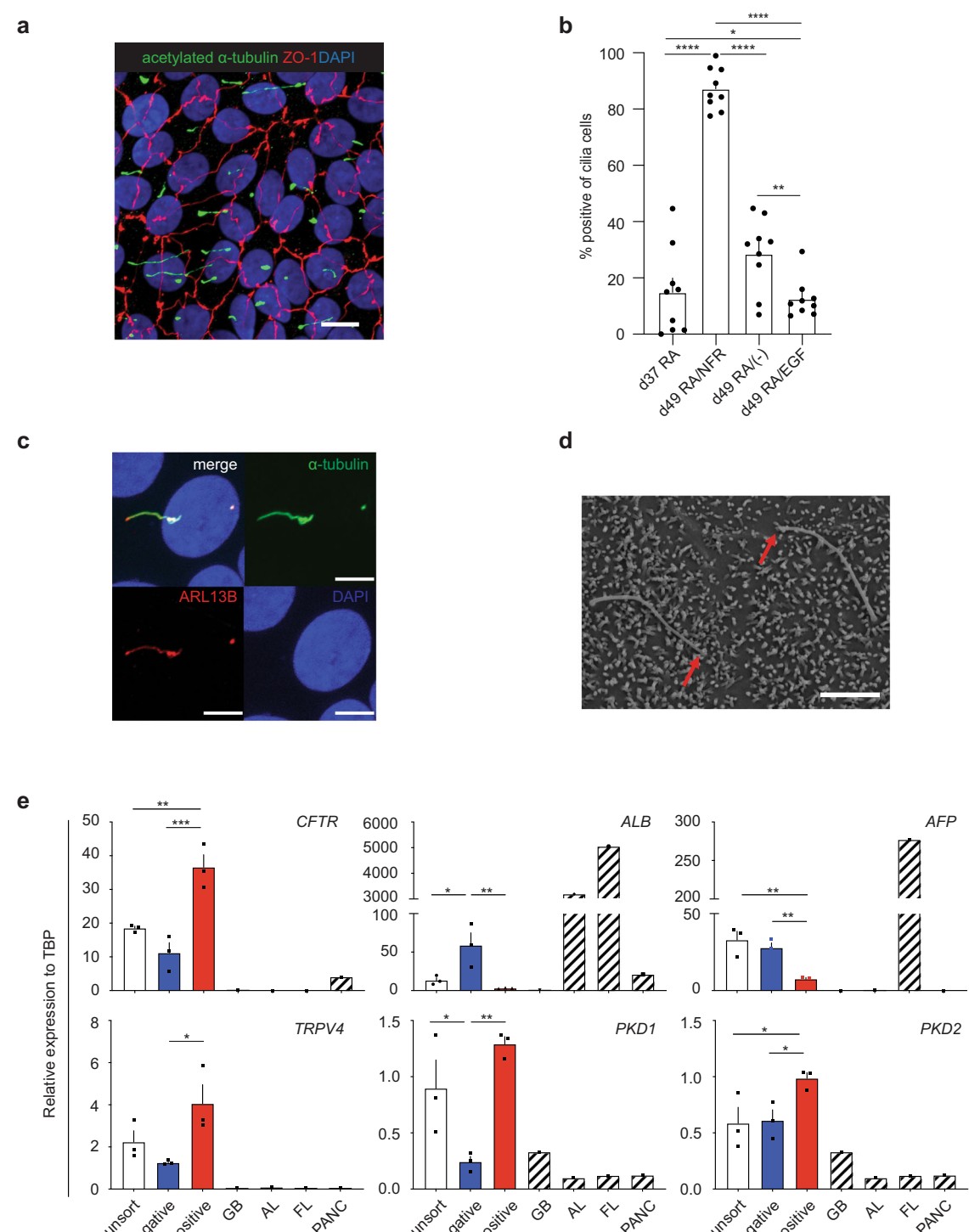

**Fig. 3 DHIC5-4D9 positive cells show primary cilia, a characteristic feature of cholangiocyte. a** Confocal microscopy analysis showing co-expression of acetylated α-tubulin and ZO1 in day 49 cholangiocyte. Scale bar represents 10 μm. **b** Quantification of cilia positive cells in day 37 (after RA) and day 49 hPSC-derived cholangiocytes after treatment with different indicated cytokine molecule combinations. $^*p \leq 0.05$, $^{**}p \leq 0.01$, $^{****}p \leq 0.0001$ one-way ANOVA. Data are represented as mean ± SEM ($n = 3$). **c** High magnification image with confocal microscopy demonstrating co-localization of ARL13B with acetylated-α-tubulin (+) primary cilia. Scale bar represents 5 μm. **d** Scanning electron microscope image of primary cilia and microvilli in 2 cholangiocytes. Red arrows indicate the tips of cilia. Scale bar represents 5 μm. **e** qPCR analysis for indicated genes in flow-based sorting fractions of DHIC5-4D9+ cells in day 49 cholangiocyte. GB gall bladder, AL adult liver, FL fetal liver, PANC pancreas. $^*p \leq 0.05$, $^{**}p \leq 0.01$, $^{***}p \leq 0.001$ one-way ANOVA. Data are represented as mean ± SEM ($n = 3$).

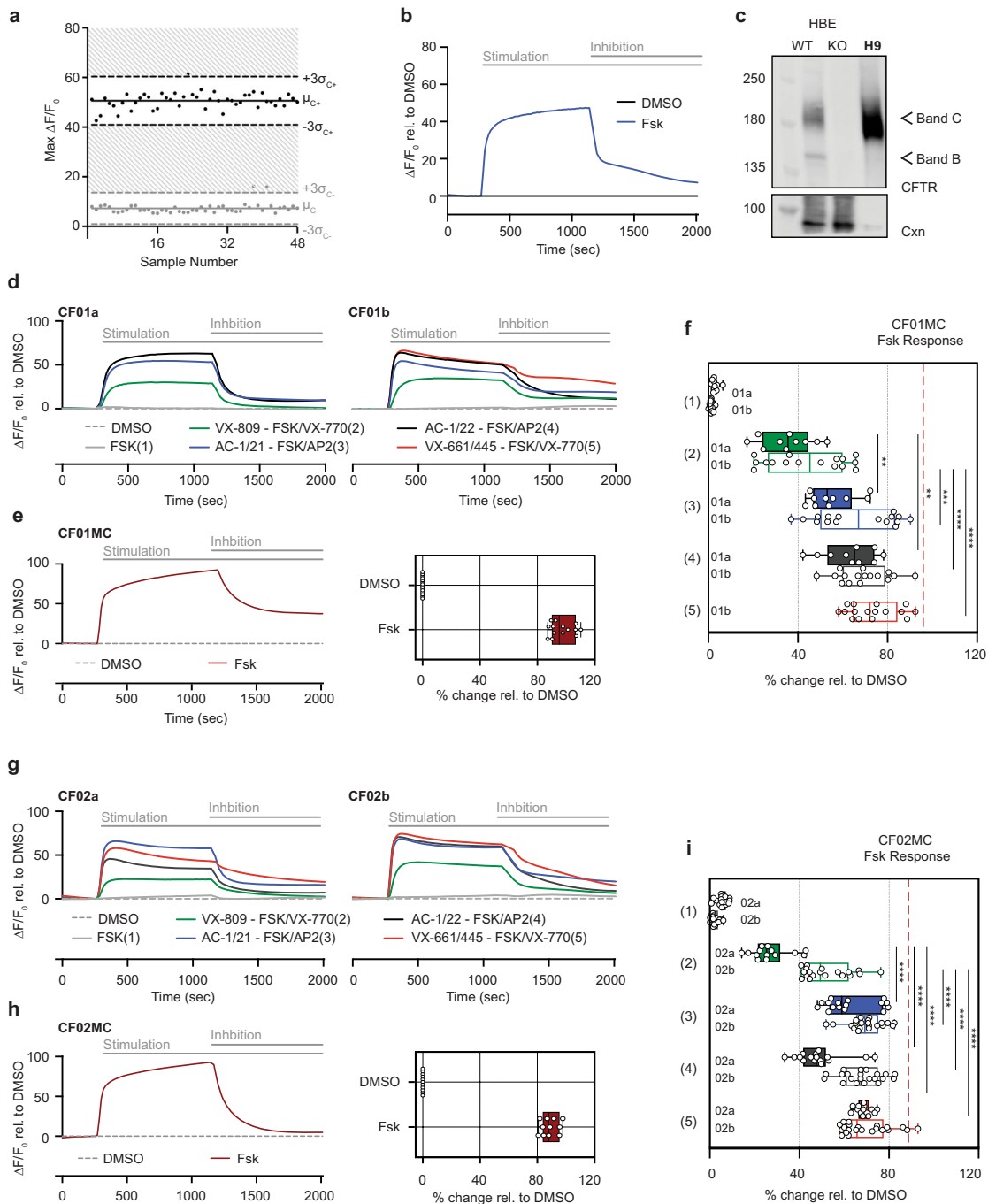

(CF01MC) was significantly higher than that of the mutant cholangiocytes in the absence of added modulators, consistent with the presence of functional CFTR protein. The mutant cysts showed a positive swelling response to the combination of modulators. As expected, the corrected cells showed no change in response with the addition of the drug combination (AC1/AC2-2 + AP2) (Fig. 5f). The corrected CF01 line expressed mature glycosylated CFTR protein as shown by western blotting (Supplementary Fig. 12a, c). Taken together, these findings show that the cysts generated from the cholangiocyte monolayer population do show a measurable CFTR-mediated swelling response and therefore can be used together with the FLIPR assay to assess the efficacy of new CFTR modulators designed to treat CF.

**Regulation of calcium signaling in the hPSC-derived cholangiocytes**. As a further assessment of maturation, we next evaluated intracellular calcium signaling in the cholangiocytes generated from this protocol. Intracellular calcium is released in cholangiocytes in response to hormonal signaling, extracellular ATP and cilia-mediated mechano-sensing[7]. The effect of ATP is important as its presence in the bile activates P2Y receptors localized at the apical membrane of cholangiocytes. Binding of ATP to P2Y receptors promotes an InsP3-dependent calcium release from intracellular stores, leading to calcium-dependent chloride efflux and bicarbonate secretion[40–43]. Q-RT-PCR analysis showed that the expression of genes related to calcium-induced fluid secretion including *P2YR1*, *ITPR3*, and *TMEM16A* were upregulated, in both the day 49 monolayer cells and the 3D cholangiocyte cysts compared to hepatoblasts (Supplementary Fig. 13).

**Fig. 4 CFTR Modulator profiling of CF patient-derived cholangiocytes in 96-well monolayer culture. a** The Z factors show the quality of the FLIPR assay measuring Apical Chloride Conductance (ACC) in 96-well plate with hPSC-derived cholangiocytes at day 49. **b** The kinetics of ACC by FLIPR assay in H9-derived cholangiocytes at day 49. **c** Western blotting shows the mature glycosylated CFTR band in day 49 H9 cholangiocyte along with the controls from CFTR expressing human bronchial epithelial HBE cell line (WT wild type, KO CFTR knock out). CNX calnexin control. Uncropped blots in Source Data. **d** The kinetics of ACC by FLIPR assay among different drug treatment in CF01a (left) and CF01b (right) iPSC-derived cholangiocytes. **e** The kinetics of ACC by FLIPR assay in CF01MC iPSC-derived cholangiocytes (left). Representative % value of CFTR channel activity with an exposure of DMSO and FSK in CF01MC iPSC-derived cholangiocytes normalized to DMSO ($n = 4$, right). **f** Representative % value of CFTR channel activity with an exposure of DMSO and CFTR modulators in CF01 iPSC-derived cholangiocytes normalized to DMSO. Dashed line indicates FSK response in CF01MC-derived cholangiocytes. Solid and open boxes represent clone a and b respectively. Stars show statistical significance to VX809/VX770, one-way ANOVA (CF01a $n = 3$, CF01b $n = 4$). **g** The kinetics of ACC by FLIPR assay among different drug treatment in CF02a (left) and CF02b (right) iPSC-derived cholangiocytes. **h** The kinetics of ACC by FLIPR assay in CF02MC iPSC-derived cholangiocytes (left). Representative % value of CFTR channel activity with an exposure of DMSO and FSK in CF02MC iPSC-derived cholangiocytes normalized to DMSO ($n = 4$, right). **i** Representative % value of CFTR channel activity with an exposure of DMSO and CFTR modulators in CF02 iPSC-derived cholangiocytes normalized to DMSO. Dashed line indicates FSK response in CF02MC-derived cholangiocytes. Solid and open boxes represent clone **a** and **b** respectively. Stars show statistical significance to VX809/VX770, one-way ANOVA (CF02a $n = 4$, CF02b $n = 4$). $^*p \leq 0.05$, $^{**}p \leq 0.01$, $^{***}p \leq 0.001$, $^{****}p \leq 0.0001$, (Median ± SEM), Box plots include median line and IQR ranges at the box bounds with the whiskers defining the minima and maxima data points.

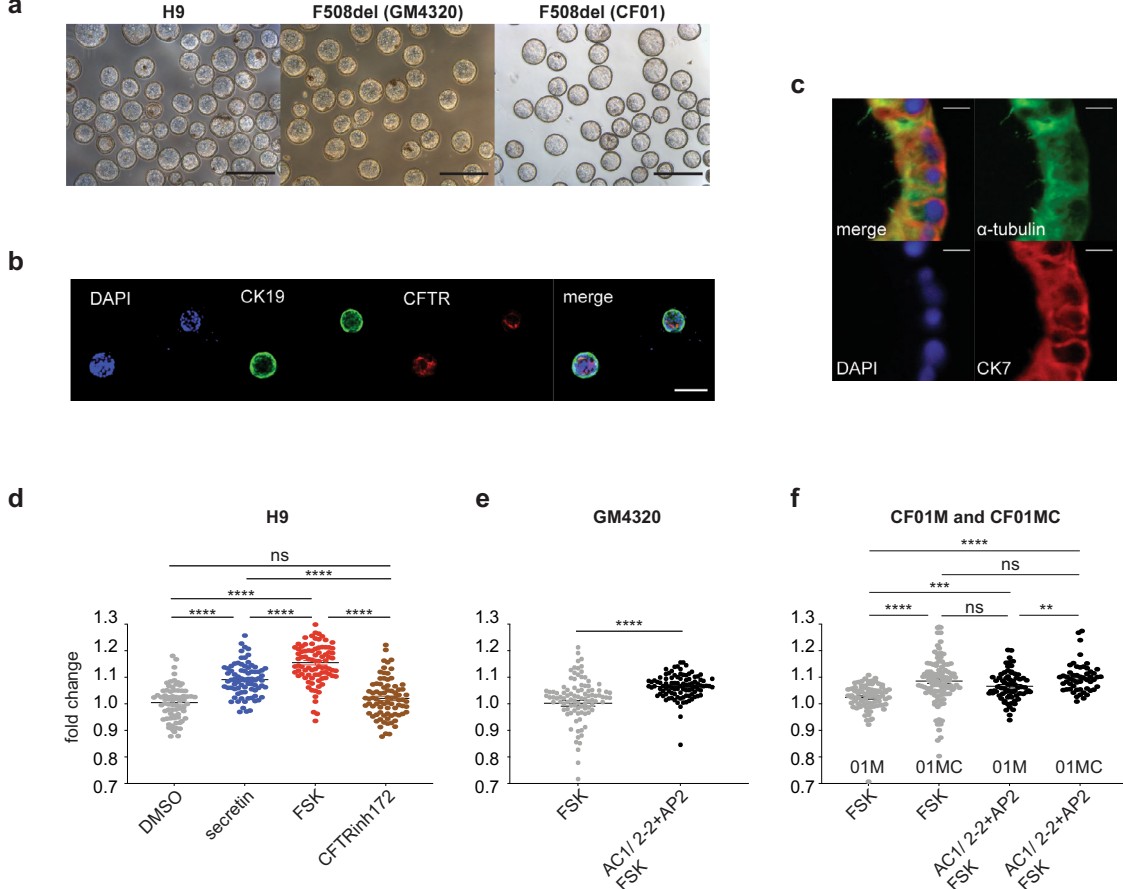

**Fig. 5 3D cholangiocytes cyst/organoids show CFTR function and modeling of CFLD. a** Photomicrographs of cyst/organoid structures that develop in liquid culture condition from monolayer hPSC-derived cholangiocytes. Scale bar represents 500 μm. **b** Microscopic analysis showing the co-expression of CK19 (green) and CFTR (red) in H9-derived cholangiocyte cysts. Scale bar represents 100 μm. **c** Immunostaining analysis showing the co-expression of acetylated-α-tubulin (green) and CK7 (red) in H9-derived cholangiocyte cysts. Scale bar represents 10 μm. **d** Quantification of the degree of H9-derived 3D cholangiocyte cyst swelling 1 h after DMSO and secretin. FSK stimulation in the absence or presence of CFTR inhibitor CFTR Inh-172. $^{****}p \leq 0.0001$ one-way ANOVA ($n = 4$ biologically independent experiments from DMSO and secretin, $n = 3$ biologically independent experiments from FSK and CFTRinh172). **e** Quantification of the degree of F508del (GM4320)-derived 3D cholangiocyte cyst swelling 1 h after FSK stimulation in the presence of DMSO or CFTR modulators. $^{****}p \leq 0.0001$ two-tailed Student's $t$ test ($n = 3$). **f** Quantification of the degree of F508del (CF01) or corrected F508del (CF01MC)-derived 3D cholangiocyte cyst swelling 1 h after FSK stimulation in the presence of DMSO or CFTR modulators. $^{**}p \leq 0.01$, $^{***}p \leq 0.001$, $^{****}p \leq 0.0001$, one-way ANOVA ($n = 3$). Data are represented as mean ± SEM. Each dot represents cyst and $n = 15–40$ cysts were measured per microwell.

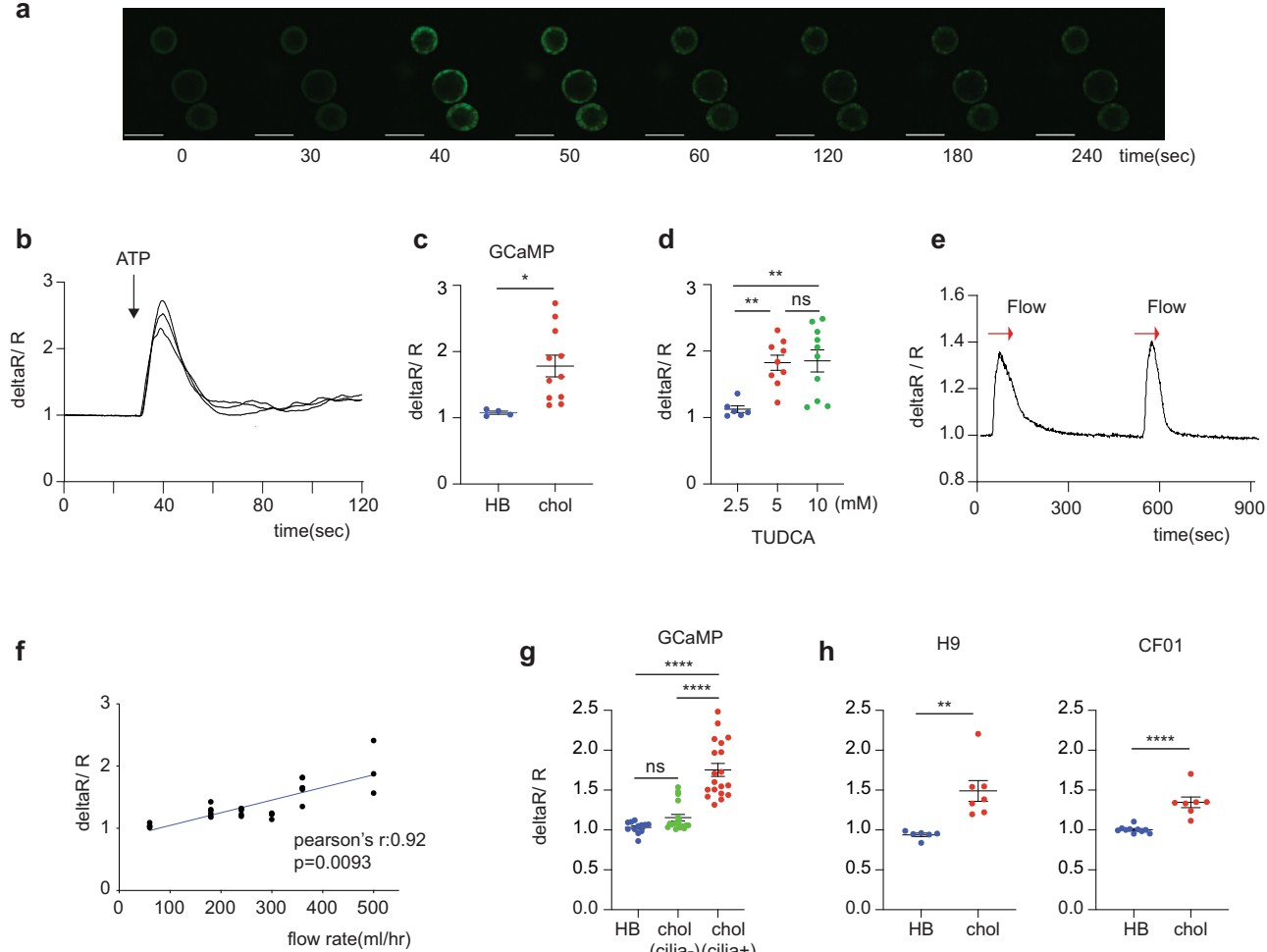

**Fig. 6 Intracellular Calcium release in ciliated hPSCs derived cholangiocyte. a** Representative time lapse image of calcium influx in GCaMP hES derived cholangiocyte cyst (3D) in response to ATP. Scale bar represents 200 μm. **b** Representative traces from Fig. 6a showing the intra cellular calcium release in GCaMP hESC-derived cholangiocytes in the response to ATP. **c** Quantification of maximum fluorescent intensity representing intra cellular $Ca^{2+}$ release in GCaMP hESC-derived hepatoblasts (HB) and cholangiocytes (chol) in the response to ATP, $^*p \leq 0.05$ two-tailed Student's $t$ test ($n = 4$ biologically independent samples from HB, $n = 3$ biologically independent samples from chol). **d** Quantification of maximum fluorescent intensity representing intra cellular $Ca^{2+}$ release in GCaMP hESC-derived cholangiocytes in the response to different concentrations of TUDCA, $^{**}p \leq 0.01$, one-way ANOVA ($n = 3$ biologically independent experiments per each concentration). Data are represented as mean ± SEM. **e** Representative trace of the intra cellular calcium release in GCaMP hESC-derived cholangiocytes induced by flow in the absence of ATP. **f** Quantification of maximum fluorescent intensity in GCaMP hESC-CF01derived cholangiocytes with increased flow rate ($n = 3$). Two-tailed pearson correlation coefficient was calculated by Prism8. **g** Quantification of maximum fluorescent intensity representing intra cellular $Ca^{2+}$ release in indicated GCaMP derived hepatoblasts (HB), non-ciliated cholangiocyte (cilia-), and ciliated cholangiocytes (cilia + ) in the response to flow, $^{****}p \leq 0.0001$ one-way ANOVA ($n = 3$ biologically independent experiments per each group). **h** Quantification of maximum fluorescent intensity representing intra cellular $Ca^{2+}$ release in indicated hPSC-derived hepatoblasts (HB) and cholangiocytes (chol) in the response to flow, $^{**}p \leq 0.01$, $^{****}p \leq 0.0001$ two-tailed Student's $t$ test ($n = 3$ biologically independent experiments per each group). Each circle represents the structure measured for intracellular calcium release.

To monitor calcium release in real time, we generated 3D cholangiocyte cysts from a hPSC line GCaMP3[44], which contains a GFP fluorescent reporter targeted to calmodulin and M13 domain from myosin light chain kinase. This reporter line enables the detection of intracellular calcium release by upregulation of GFP[45–48]. Addition of ATP to the GCaMP3-derived cholangiocyte cysts induced fluorescent transients indicative of intracellular calcium release (Fig. 6a–c, Supplementary Figure 14a, and Supplementary Video 4). Aggregates of hepatoblasts which do not express P2Y1 and ITPR3 did not show this activity (Fig. 6c). Bile acids are known to stimulate fluid secretion from cholangiocyte through calcium-activated chloride channels[43]. When the GCaMP3-derived cholangiocytes were exposed to Tauroursodeoxycholic acid (TUDCA), calcium release was activated in dose-dependent manner (Fig. 6d and Supplementary Video 5). An ATP-induced GFP response was also measurable in the cyst-derived cholangiocytes following adherence of the cells to fibronectin coated plastic indicating both 3D and 2D formats are applicable for the functional studies (Supplementary Fig. 14b–e and Supplementary Video 6). Under these conditions, the cysts attach and form a monolayer on the bottom of the chamber. The attached cholangiocytes from the cysts also contained primary cilia as shown by immunostaining against acetylated α- tubulin (Supplementary Fig. 14f). Next, we tested if flow could induce calcium release, mimicking the reaction of the primary cilia to bile flow in the liver. For these studies the 3D cholangiocyte cysts were plated in a flow chamber and media was moved over these cells using a Peristaltic Bio Mini Pump. As shown in

Supplementary Fig. 14g, fluid flow induced a fluorescent signal indicating that the movement of the media over the surface of the cholangiocytes resulted in the release of intracellular calcium (Supplementary Fig. 14h, i and Supplementary Video 7). Following the initial response, intracellular calcium release was observed with an additional flow stimulus (Fig. 6e). The magnitude of the flow-induced response increased with flow rates (Fig. 6f). This flow induced calcium response was not observed in cholangiocytes treated with EGF that inhibits primary cilia formation (Fig. 6g, Supplementary Fig. 14j, and Supplementary Video 8). We also confirmed similar intracellular calcium release in H9 and F508 del CF iPSCs-derived cholangiocytes in response to flow by Fluo-4 (Fig. 6h). These findings demonstrate that the hPSC-derived cholangiocytes are able to release intracellular calcium in response to ATP and fluid flow, two pathways that regulate calcium signaling in cholangiocytes in vivo.

**Flow stimulation induces CFTR functional activity via calcium signaling in hPSC-derived cholangiocytes.** Previous studies have shown that the bending of cholangiocyte cilia by luminal fluid flow in intrahepatic rat bile duct explants induced an increase in calcium and cAMP signaling, demonstrating a link between the mechanosensing properties of the cilia and intracellular signaling[3,6]. To examine the status of mechanical stimulation of the hPSC-derived cholangiocytes, we first determined if fluid flow could induce cilia bending in these cells. As shown in Fig. 7a, the cilia in the cholangiocytes did bend when exposed to flow rate of 102 µl/sec. To assess the consequence of cilia bending on the physiology of the cholangiocytes, we measured calcium signaling and apical chloride conductance (ACC) in GCaMP3 hPSC-derived cells subjected to fluid flow. As shown in Fig. 7b, the uptake of calcium signaling, and ACC activity was induced by fluid flow stimulation. Interestingly, the addition of thapsigargin, a potent inhibitor of sarco endoplasmic reticulum $Ca^{2+}$-ATPase that leads to a depletion of ER calcium storage, diminished fluid flow mediated increases in both calcium signaling and ACC activity (Fig. 7c). These findings suggest that fluid flow mediated increases in cytosolic calcium are functionally correlated to chloride channel activity (Fig. 7d). The flow-induced ACC response in H9-derived cholangiocytes were further augmented by the addition of FSK, known to activate CFTR channels through protein kinase A phosphorylation. The additive effect of flow and FSK increased the ACC response to level observed in the monolayer cultures (Fig. 7e, f). Flow induced ACC was not detected in CF patient iPSC-derived cholangiocytes. However, CFTR function was rescued in the patient cholangiocytes by treatment with the corrected compound, VX809 and significantly higher levels of rescue were observed by addition of the AC-1/AC2-2 drug combination of correctors in the presence of flow (Fig. 7g, h). Taken together, these findings suggest that flow stimulation increases calcium signaling which in turn regulates CFTR channel activity and controls fluid secretion.

**Global transcriptomic analyses of cholangiocytes from monolayer cultures and 3D cysts.** To further characterize the cholangiocytes from the monolayers and the 3D cysts, we carried out single-cell RNA sequencing analyses on the two populations. Following filtering steps and batch effect correction based on the detection of the mutual nearest neighbors (MNNs), a total of 4648 monolayer cells (2D) and 3447 cells from the cysts (3D) were analyzed. Clustering of the combined data set identified 12 distinct populations (Fig. 8a, b and Supplementary Data 1). The top five differentially expressed genes among 12 distinct clusters are shown in Supplementary Fig. 15a. The expression of SOX9, SOX4,

ONECUT1, ONECUT2 (HNF6b), EPCAM, CD24, and KRT19 suggests that they represent cholangiocytes (Fig. 8c and Supplementary Fig. 15b). Small subsets of clusters 8, 9, 10, and 11 express secretory protein and hormonal genes suggesting the contamination of gastrointestinal (GI) secretory cells. Cluster 5, the most prominent in the 3D cysts contains cells that express genes indicative of hepatocyte/hepatic progenitor cells. Cluster 7 was annotated as proliferating cholangiocytes given its high G2/M cell cycle score and expression of BRCA2, CDC6 and CDK1 in both monolayer and cyst cells (Fig. 8d, Supplementary Fig. 15c, and Supplementary Table 1). Both monolayer and cyst populations expressed the core functional transporter genes CFTR, SLC4A2, SCTR (Fig. 8c), and primary cilia related genes (Supplementary Fig. 15d). To establish the identity of the hPSC-derived cholangiocytes, we investigated their expression patterns of gene signatures known in human intrahepatic bile ducts (IHBD), extra hepatic ducts (EHBD), and gall bladder (GB)[49]. As shown in Fig. 8e, our cholangiocytes expressed genes associated with IHBDs (SOX4, BICC1, DCDC2, PKHD1, and JAG1) but not those indicative of EHBD nor GB identity suggesting that they represent the subset of cholangiocytes that form the bile ducts within the liver. To further characterize the molecular profiles of cholangiocytes generated in the monolayer cultures and 3D cysts, we analyzed differentially expressed genes in three distinct populations; a monolayer dominant population (2D) from clusters 4 and 6 made up of 97.9% cells from the monolayer, 2.1% from cysts, a cysts dominant population (3D) from cluster 2 and 3 consisting of 29.3% cells from the monolayer and 70.9 % from cysts, and an combine population from clusters 0 and 1 comprised of 71.6% cells from the monolayer and 28.4% from cysts (Fig. 8b, Supplementary Table 2, and Supplementary Data 2). Top 10 differentially expressed genes (ranked by logFC) of the three populations are shown in Supplementary Fig. 15e. Clusters 5 (progenitor cells), 7 (proliferative cells), and 8–11 (small subsets of GI secretory cells) were not included in this analysis. Importantly, we identified an enrichment of AQP1 in the 3D dominant population that was consistent with our qPCR analysis that shown in Supplementary Fig. 13. Gene ontology analysis based on the upregulated genes revealed that endoplasmic reticulum (ER) related signal pathways were significantly enriched in the cysts, whereas the cell junction and extracellular matrix organization related pathways were higher in monolayer cells (Supplementary Fig. 15f). We next compared the transcriptional profiles of the hPSCs derived cholangiocytes to those of populations found in the primary human liver scRNA-seq analyses[50] by following the mutual nearest neighbors batch correction method (Supplementary Fig. 16a–c). Notably, integration analyses of these datasets uncovered that the 3D dominant population demonstrated close similarity to the human cholangiocyte scRNA profile (Fig. 8f, g and Supplementary Fig. 16d). Pearson correlation analysis using the expression levels of the 2000 variable features further showed that the transcriptional profile of human cholangiocytes have the highest level of correlation to that of the 3D dominant population compared to the 2D dominant and combine populations (Fig. 8h and Supplementary Table 3). Taken together, the findings from this molecular analysis demonstrates that the hPSC-derived cholangiocytes in 3D cyst organoid shows similar molecular profiles to those in intrahepatic cholangiocyte from human adult liver tissue.

**Transplantation of hPSC-derived cholangiocytes.** The ability to generate functional cholangiocytes from hPSCs raises the interesting possibility of developing cell-based therapies to regenerate deficient and/or diseased bile ducts in patients suffering from cholangiopathies. For this approach to be feasible, it is necessary to demonstrate that transplanted cholangiocytes can colonize the

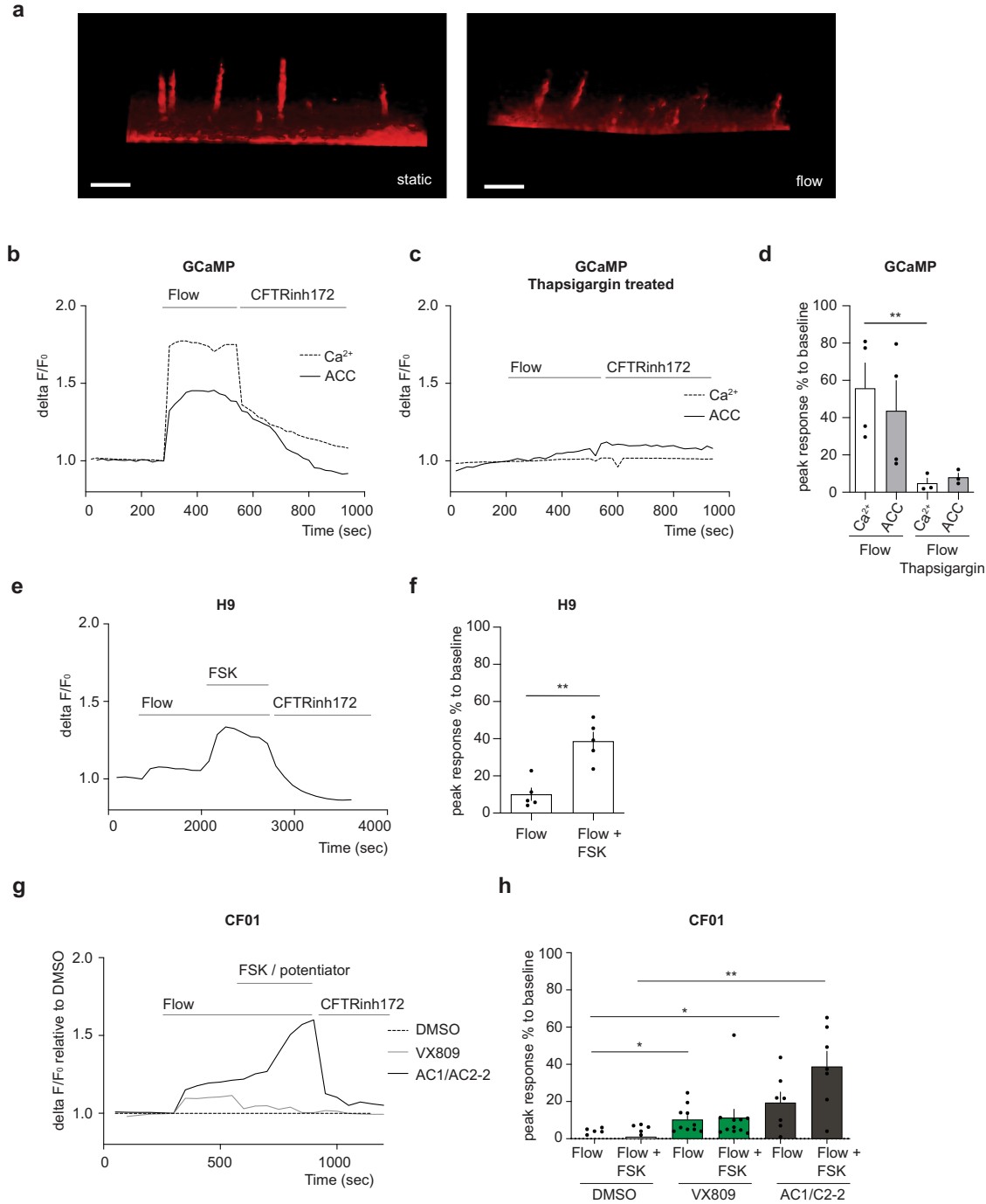

**Fig. 7 CFTR function in ciliated hPSC-derived cholangiocyte in response to flow. a** Microscopic images show that cilium extend straight from the cell surface in the static condition (left). Flow (102 μl/s) bent cilium (right). Scale bars represent 5 μm. **b** Correlation between calcium influx and CFTR function. Continuous flow induced calcium influx and Apical Chloride Conductance (ACC) in GCaMP derived cholangiocyte. **c** Representative trace of the Apical Chloride Conductance (ACC) and calcium influx in GCaMP derived cholangiocyte with the presence of thapsigargin in response to flow. **d** Quantitative analysis showing the peak response of calcium and ACC, $^{**}p \leq 0.01$ one-way ANOVA. Data are represented as mean ± SEM ($n = 3$–4). **e** Representative trace of the Apical Chloride Conductance (ACC) in H9-derived cholangiocyte cysts in response to flow followed by flow in the presence of FSK. Response was blocked by CFTR inhibitor172. **f** Quantitative analysis showing the peak response of ACC in H9-derived cholangiocyte cysts in response to flow, $^{**}p \leq 0.01$ two-tailed Student's *t* test ($n = 5$). **g** Representative trace of the apical chloride conductance (ACC) in CF01 derived cholangiocyte after treated with CF modulators in response to flow. **h** Quantitative analysis showing the peak response of ACC in CF01 derived cholangiocyte in response to flow in the presence of CFTR modulators, $^{*}p \leq 0.05$, $^{**}p \leq 0.01$ one-way ANOVA. Data are represented as mean ± SEM ($n = 5$–11).

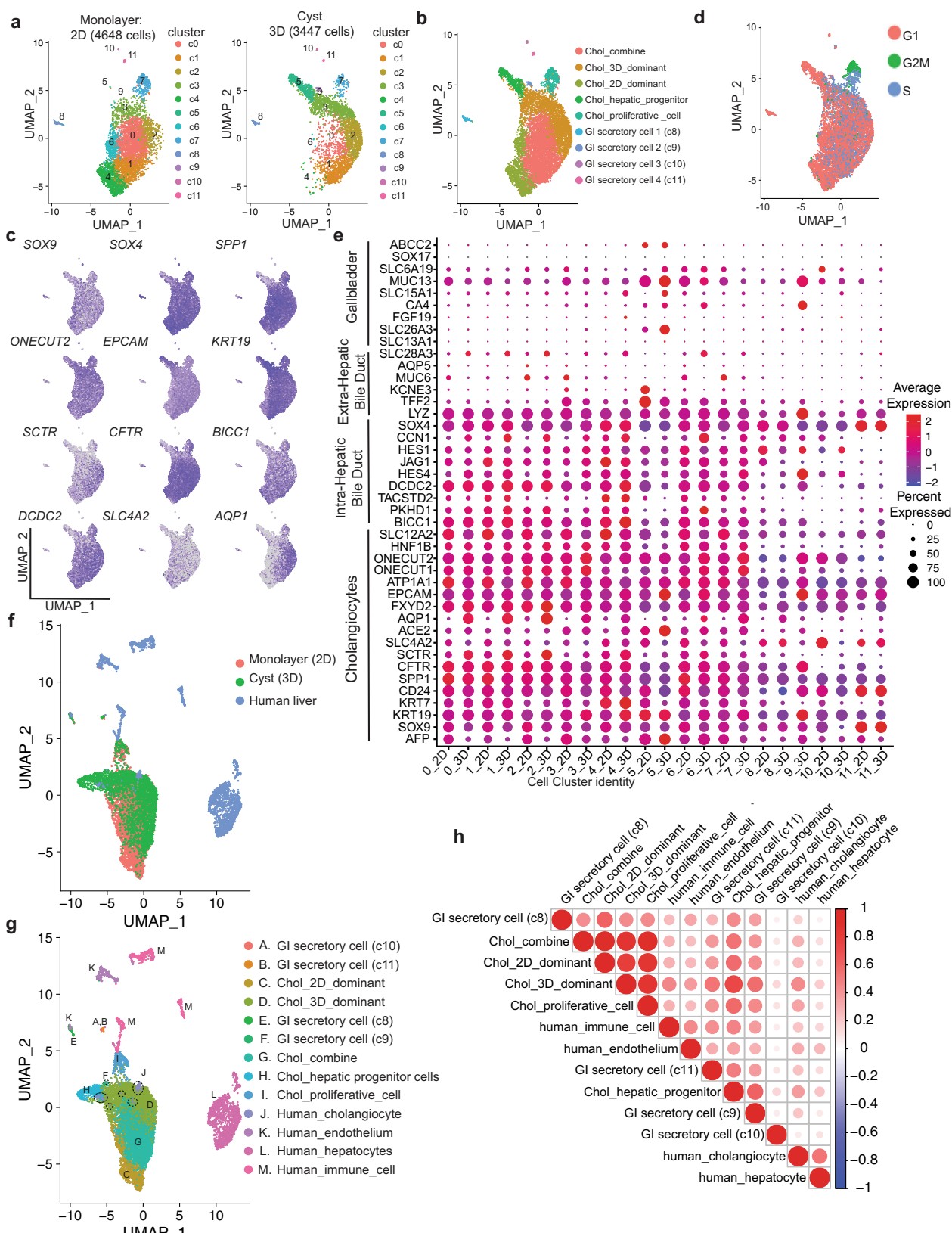

liver of a recipient and generate new ductal structures. To test this, we transplanted $10^6$ day 55 cholangiocytes into the spleen of TK-NOG mice[51]. Six weeks following transplantation, multiple duct structures consisting of cells that expressed a human mitochondria gene were detected throughout the liver (Fig. 9a). The cells within the ducts also expressed CK7, CK19, and contained cilia (Fig. 9b). The human CK19 positive cells were aligned with mouse CK19 cells to form the luminal structure in the liver suggesting that they may be integrated into the mouse biliary system (Fig. 9c). As a parallel approach, we transplanted $10^6$ the mature cholangiocytes (day 55) into the subcapsular space of the kidney in NSG mice. Four weeks following transplantation,

**Fig. 8 Global gene analysis of 2D and 3D cyst/organoids hPSC-derived cholangiocytes. a** UMAP projection of 2D (left) and 3D (right) hPSC-derived cholangiocytes labeled by cell cluster. **b** UMAP projection of 2D dominant, 3D dominant, and 2D/3D combine populations of hPSC-derived cholangiocytes. **c** UMAP plots displaying the expressions of selected cholangiocyte markers. **d** UMAP plots showing cells in various cell cycles. **e** Dot plots displaying expression of genes indicative of cholangiocytes and known markers for different regions of primary bile ducts. The plot is split by cluster ID (0–11) and data source (2D vs. 3D). Size of the dot represents proportion of the population that expresses each gene. Color indicates the means of average expression level. **f** UMAP projection of 2D and 3D hPSC-derived cholangiocytes and adult human liver cells labeled by data source. **g** UMAP projection of 2D and 3D hPSC-derived cholangiocytes and human adult liver cells labeled by pre-annotated cell type. Circles with dotted line represent human adult cholangiocyte (cluster J). **h** Correlation matrix summarizing Pearson correlation coefficient measuring the correlation between hPSC-derived cholangiocyte populations and human adult liver cells.

multiple duct structures that express CK7, human mitochondria, and α- acetylated tubulin were also detected (Fig. 9d, e). Ter-atomas were not observed in any of the mice that received transplants. Collectively these findings demonstrate the feasibility of generating ductal structures in the liver and ectopic sites of immunocompromised mice followed transplantation of hPSC-derived cholangiocytes.

## Discussion
The utility of hPSC-derived cholangiocytes for therapeutic applications is dependent on the capacity of the cells to recapitulate the key functions of mature bile duct cells in vitro. To derive appropriate cells, it is necessary to recapitulate in the differentiation cultures the critical developmental stages and regulatory events that control their specification, expansion and functional maturation in the fetus and neonate. In our previous study, we generated hPSC-derived cholangiocytes organoid structures in semi-solid media in an attempt to recapitulate the 3D structure of the developing liver[21]. While effective, this approach has several drawbacks including inaccessibility of cells and heterogeneity of populations within the organoids. Other studies have described both 3D and 2D culture systems that promote the development of hPSC-derived cholangiocytes that display some characteristics of functional bile duct cells. None of these protocols, including the one we described give rise to functional ciliated cells[20–25]. To address this issue, we designed a monolayer-based strategy that enabled the identification of signaling molecules that promote efficient maturation, yielding populations that contain a high proportion of functional ciliated cholangiocytes. Furthermore, cholangiocyte cysts were efficiently generated from the monolayer. These cholangiocytes showed many characteristics of functional biliary cells including the presence of CFTR channels, the ability to mobilize calcium, responsiveness to cilia-mediated mechano-sensing, and the capacity to engraft the liver of immunocompromised animals.

Previous studies in mice and the hPSC differentiation model have shown that Notch signaling plays a pivotal role in specification of the cholangiocyte fate from hepatoblasts[52,53]. Through the use of two different screening strategies in this study, we were able to identify the retinoic acid, BMP, cAMP, and Rho kinase pathways as additional regulators of human cholangiocyte development and maturation. These pathways are not redundant but rather appear to function in a stage specific pattern that establishes the framework of a regulatory roadmap of the human cholangiocyte lineage. The first step in this process is the induction of the cholangiocyte fate from hepatoblasts by Notch signaling. Patterning of the Notch-induced progenitors by RA signaling promotes the development of immature CFTR+ cholangiocytes that in response to manipulation of the NOG/FSK/RI pathways give rise to ciliated, functional DHIC5-4D9+ cholangiocytes. Although a role for these pathways has not been well defined in cholangiocyte development in vivo, recent studies showing that RA signaling impacts CFTR function in mouse spermatogonia stem cells[54] supports our observations in the

hPSC-derived cells. The observation that the addition of EGF reduces the proportion of ciliated cholangiocytes distinguishes our study from all others published to date that include EGF in the maturation strategy[20,22–25]. The differences in the requirement for EGF may reflect the fact that our endpoint was ciliated cells, whereas others used different parameters including CK19, CK7, and SOX9 expression to measure cholangiocyte development.

Primary cilia are organelles protruding from the apical surface of many different types of epithelial cells that function to maintain tissue homeostasis by detecting changes in the extracellular environment[55,56]. The importance of cilia to normal cell and tissue function is highlighted by the fact that mutations in genes required for ciliogenesis and primary cilia function lead to disorders/diseases known as ciliopathies[57,58]. One such disease, polycystic kidney and liver disease is caused by mutations in either PKD1, PKD2, or PKHD1 and is characterized by the presence of multiple cysts in the portal triad and progressive hepatic fibrosis[59,60]. Our understanding of the role of cilia in normal cell physiology and the development of ciliopathies has been hampered by the lack of accessible model systems to study the function of these organelles. The findings presented here that flow induced calcium signaling and CFTR activation in the hPSC-derived cholangiocytes strongly suggests that the cilia are functional and capable of mechanotransduction of signals necessary for normal cell function. With these advances, it will now be possible to model ciliopathies with patient iPSCs and investigate in detail the mechanisms that lead to the onset of these diseases and identify therapeutics to treat them.

Over the past decade, significant progress has been made in the development of CFTR protein targeted drug therapies. While the most recent FDA approved CF drugs are able to enhance lung function in 90% of CF patients carrying the common F508del-CFTR mutation[61,62], there remains a number of hurdles to achieve an effective therapeutic strategy for all patients. A major issue with the drugs approved to date is that the response in different patients carrying the same mutation is variable, ranging from very modest improvement in function to an almost complete recovery of function. A second challenge is that patients with rare mutations do not respond to the drugs. Finally, the effect of the drugs is mainly measured on lung function, while the disease targets multiple organs including the pancreas, liver, and intestine. To overcome these challenges, one needs to develop assays for multiple organs from different individuals with the common F508del-CFTR mutation as well as other rare mutations. The monolayer protocol described here that gives rise to cholangiocytes with a high Z prime score provides a source of cells and culture format for measuring the effects of CF drugs on cells involved in CFLD. Indeed, we were able show that CF drugs display different degrees of rescue in these cholangiocytes generated from different patients. Additionally, with access to these mature cells, we demonstrated that it is possible to measure the effect of drugs on CFTR function in the absence of exogenous cAMP signaling, under conditions of fluid flow that were

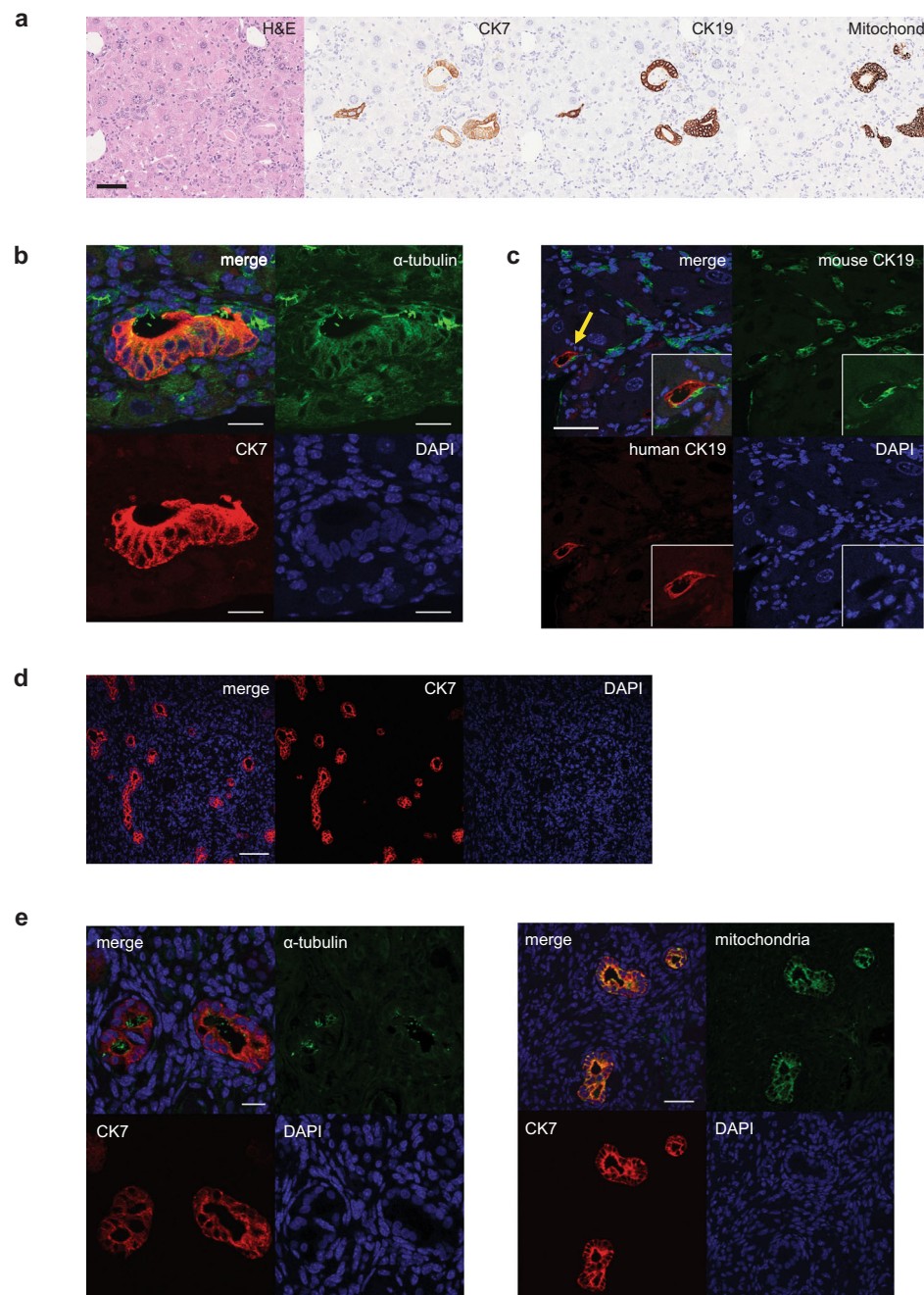

**Fig. 9 Engraftment of hPSC-derived cholangiocyte in mice liver. a** Photomicrographs of duct-like structures generated from hPSC-derived cholangiocytes in the liver of TK-NOG mouse after 6 weeks of transplantation. HE and DAB staining with human specific antibodies. Scale bar represents 50 μm. **b** Confocal image of a histological section in the liver of TK-NOG mouse after 6 weeks of transplantation showing duct structures with primary cilia (green) counterstained with CK7 (red) and DAPI (blue). Scale bar represents 20 μm. **c** Confocal image of a histological section in the liver of TK-NOG mouse after 6 weeks of transplantation co-expressing human CK19 (red) and mouse CK19 (green). Scale bar represents 50 μm. **d** Confocal image of histological section showing duct structures in the kidney of NSG mouse showing human CK7 (red). Scale bar represents 100 μm. **e** Left: high magnification image of confocal microscopy demonstrating that hPSC-derived cholangiocyte display co-localization of CK7 (red) with acetylated α-tubulin (green) in the mouse kidney subcapsular space. Scale bar represents 20 μm. Right: high magnification image of confocal microscopy demonstrating that hPSC-derived cholangiocytes display co-localization of CK7 (red) with human mitochondria (green) in the mouse kidney subcapsular space. Scale bar represents 50 μm.

designed to mimic aspects of bile flow in the bile duct. Future studies aimed at developing organ-on chip platforms to more accurately replicate fluid flow will enable us to gain a better understanding of the interaction between cilia and CFTR function.

Although cholangiocytes represent only a small portion of the liver, they are essential for maintaining hepatocyte metabolism by controlling the viscosity through modifications of bile acid that include ion carriers and various molecules[1]. Diseases that affect cholangiocyte function such as primary sclerosing cholangitis, primary biliary cirrhosis, CFLD, and biliary atresia can impact the viscosity and flow of bile acid, resulting hepatocyte toxicity[63]. Regardless of the etiology of disease, the only effective current treatment for advanced stages of these cholangiopathies is liver

transplantation. As the shortage of donor organs limits the number of patients that can be transplanted, other interventions such as cell replacement therapy need to be developed to treat patients with cholangiopathies. A recent study in mice has shown that it is possible to regenerate biliary structures from transplanted cells in the mouse model of bile duct paucity. In this case, adult hepatocytes were shown to have the potential to generate cholangiocytes and connect to the existing biliary network in the mouse liver[64]. Our findings show that following transplantation, the hPSC-derived cholangiocytes can migrate from the spleen to the liver and form the luminal ductal structures with existing bile duct epithelial cells. These findings suggest that the engrafted cells are functional in this mouse model, and support for the concept of cell-based therapy to treat cholangiopathies. This is further supported by a recent study that demonstrated the successful engraftment of human primary cholangiocytes organoids in a mouse model of bile duct injury and ex-perfusion model of human liver[49].

The finding from our single-cell RNAseq analysis showed that the majority of cells in both cholangiocytes generated in monolayer culture and 3D cysts express cholangiocyte specific genes and the transcriptional profile of our cholangiocytes was aligned well with that of intrahepatic cholangiocytes[49]. Gene ontology analysis revealed that monolayer is more supportive for ECM organization and 3D is more supportive for protein targeting suggesting the differences in culture format may reflect the difference in the state of maturation. Notably, the transcriptomics of the cholangiocytes in the 3D dominant populations display similarity to those of adult human cholangiocytes, indicating that 3D formation may facilitate further maturation of hPSC-derived cholangiocytes in vitro.

In summary, we report on the successful generation of functional ciliated cholangiocytes from hPSCs that will provide opportunities to study and treat diseases of the liver. Both monolayer- and organoid-derived cells are adaptable for further bioengineering approaches to produce more complex hPSC-derived liver tissue with biliary ductal structures. Additionally, the successful engraftment of differentiated cholangiocytes in the liver of immune-incompetent mice provides the basis for developing innovative therapeutic strategies to treat cholangiopathies.

## Methods

**Human ES and iPS cells differentiation to cholangiocytes in monolayer**. hES (H9: WA009 and GCaMP3) and iPSC cells (WT01, CF01a, CF01b, CF01MC, CF02a, CF02b, CF02MC, and GM4320) were maintained on irradiated mouse embryonic fibroblasts in hESC medium as described previously[65,66]. hPSCs were differentiated to hepatoblasts as previously described with slight modifications[21,38]. Briefly, prior to the induction of endoderm in the monolayer culture, hES/iPS cells were passaged onto 2.5% Matrigel (Corning: 354230) coated surface plates for 1 day at seeding densities of 100,000–175,000 cells per well in a 12-well culture dish (plating density varied depending on each hPSC line). To induce endoderm differentiation, cells were cultured for 1 day in RPMI 1640 medium (Themo Fisher: 11875093) supplemented with glutamine (2 mM, Themo Fisher: 25030081), MTG (4.5 ×10⁻⁴ M, Sigma: M6145), activin A (100 ng/ml, R&D: 388-AC/CF), and CHIR99021 (2 μM, TOCRIS: 4423). At day1, CHIR99021 was removed, and cells were cultured for the next 2 days in RMPI supplemented with glutamine, ascorbic acid (50 μg/ml, Sigma: A4544), MTG, basic fibroblast growth factor (bFGF, 5 ng/ml, R&D: 233-FB), activin A (100 ng/ml) followed by 4 days in serum-free-differentiation (SFD) based medium with the same supplements. The media was changed every 2 days. At day 7, the definitive endoderm, which is confirmed as positive for CXCR4 and cKIT by flow cytometry, was specified to a hepatic fate by culture in low glucose DMEM (Thermo Fisher: 11885084) containing bFGF (40 ng/ml) and Bone Morphogenic Protein (BMP4, 50 ng/ml, R&D: 314-BP) and supplemented with 1% vol/vol B27 supplement (Thermo Fisher: 17504044), ascorbic acid, glutamine, and MTG. The media was changed every 2 days from day 7 to day 13. To promote the maturation of the hepatoblast population, cells were cultured in a mixture of low glucose DMEM (Thermo Fisher: 11885084)/Ham's F12 (Corning: 10080CV) (3:1) media with 0.1% BSA, 1% vol/vol B27 supplement, ascorbic acid, glutamine, MTG, Hepatocyte Growth Factor (HGF, 20 ng/ml, R&D: 294-HGN), Dexamethasone (Dex, 40 ng/ml, BioShop: DEX002), and Oncostatin M (OSM; 20 ng/ml, R&D: 295-OM/CF) for 8 days. 1 μM CHIR99021 was added in the

culture of hES cell derived hepatoblast differentiation at this stage. The differentiation including the endoderm induction, hepatic specification, and maturation from day 0 to day 21 were maintained in a low O2 incubator (5% CO₂, 5% O₂, 90% N₂). At day 21, the cells were transferred into an ambient O2 incubator and cultured in a mixture of high glucose DMEM (Thermo Fisher: 11995065)/Ham's F12 (3:1) with 0.1% BSA, 1% vol/vol B27 supplement, ascorbic acid, glutamine, MTG, HGF (20 ng/ml), Dex (40 ng/ml), and OSM (20 ng/ml) for 6 days. 30 Gray irradiated OP9 jagged-1 (OP9j) cells were prepared on 2.5% Matrigel coated wells (12 well plates) at a concentration of 200,000 cells per well of 12-well plate in alpha-modified minimum essential media (a-MEM) supplemented with glutamine and 20% fetal bovine serum at day 26. To induce cholangiocytes differentiation, day 27 hepatoblasts were dissociated by TrypLE and plated onto the irradiated OP9j cells. The plated cells were cultured in DMEM/Ham's F12 (1:1) (Corning: 10092CV) supplemented with 0.1% BSA, 1% vol/vol B27 supplement, ascorbic acid, glutamine, MTG, HGF (20 ng/ml), and Epidermal growth factor (EGF, 50 ng/ml, R&D: 236-EG) for 4 days. To induced CFTR expression in cholangiocyte-like cells, following HGF and EGF treatment, the medium was switched into DMEM/F12 medium with 0.1% BSA, 1% vol/vol B27 supplement, ascorbic acid, glutamine, MTG, and Retinoic Acid (1 μM, Sigma: R2625) for another 6 days. To promote the maturation of cholangiocytes that express primary cilia and DHIC5-4D9, the cells were cultured with DMEM/F12 medium with 0.1% BSA, 1% vol/vol B27 supplement, ascorbic acid, glutamine, MTG, Noggin (50 ng/ml, R&D: 3344-NG), ROCK inhibitor Y-27632 (5 μM, TOCRIS: 1254), and Forskolin (FSK, 5μM, TOCRIS: 1099) for 12 days. For iPSCs, day 27–33 hepatoblasts were differentiated to cholangiocytes due to the slow kinetics measured by flow cytometric ALB positivity. More than 85% of ALB positive hepatoblasts were used for the further cholangiocyte differentiation. The medium for all steps of cholangiocytes differentiation were changed every two days. The differentiation was maintained in an ambient O₂ incubator.

**Generation of 3D cholangiocyte organoids**. Day 49 cholangiocyte following the differentiation in monolayer were dissociated with collagenase type I and then small clumps of cholangiocyte cells were plated on low attachment cluster dishes and cultured with the same medium for monolayer differentiation. The 3D cholangiocyte organoids spontaneously formed cyst-like structures within 6 days. The differentiation was maintained in an ambient O₂ incubator.

**WT iPSC, CF patient iPSCs, and mutation corrected iPSCs**. iPSC lines were obtained from the CFIT Program (https://lab.research.sickkids.ca/CFIT/[67], Sick-Kids Research Ethics Board, Approval#44783). Two CF lines with 2 sub-clones (CF01a and 01b and CF02a and 02b) are carrying the common CF mutation F508del. CF01MC and CF02MC are the mutation corrected iPSCs, generated with the CRISPR-Cas9 technology from the CF01 and CF02. WT01, CF01, CF02, CF01MC, and CF02MC were generated at the Centre for Commercialization of Regeneration Medicine (CCRM) in Toronto. Authentication Information is supplied in Supplementary Information.

**Flow cytometry**. Differentiated cells were dissociated into single-cell suspensions. Dead cells were excluded during flow cytometry analyses and gating was determined using isotype control. For cell surface marker analyses, staining was carried out in PBS with 10% FCS. For detection of intracellular proteins, staining was performed on cells fixed with 4% paraformaldehyde (PFA: Electron Microscopy Science, Hatfield, PA, USA) in PBS. Cells were permeabilized with 90% ice-cold methanol for 20 min for ALB, AFP, and CK7. Cells were subsequently incubated with secondary antibodies for 30 min at room temperature. The stained cells were analyzed using LSR Fortessa flow cytometer (BD). Data were analyzed using FlowJo software 10 (Tree Star). FACS antibodies and their dilution ratio are listed in Supplementary Tables 4 and 5.

**Immunostaining**. For staining of monolayers, cells were fixed with 4% PFA for 15 min and permeabilized with 0.2% Triton X-100 or cold 100% methanol (CFTR, ASBT) before blocking. Cells were washed three times with PBS for 10 min at room temperature (RT) before and after each staining step. All antibodies for monolayer were diluted in DPBS + 0.1%BSA + 0.1% TritonX-100. For the staining of the 3D cyst structures, samples were fixed with 4% PFA and washed with DPBS and permeabilized with 0.2% Triton X-100 or cold 100% methanol (CFTR) before blocking. Cells were washed three times with PBS for 10 min at RT before and after each staining step. Antibodies for 3D cysts were diluted in DPBS + 0.3%BSA + 0.3% TritonX-100. DAPI was used to counterstain the nuclei. Monolayer and 3D cysts staining were visualized using a fluorescent microscope (Leica CTR 6000) and images were captured using Leica Application software. For the detection of primary cilia/ ZO-1 in monolayer cultures, the differentiation was carried out on Matrigel-coated coverslips, and images were visualized using the NIKON A1 Resonant Confocal Microscope and images were captured using the Nikon Elements software. Paraffin-embedded sections were dewaxed with xylene, rehydrated, placed in Tris-EGTA-buffer (TEG: 10 mM Tris, 0.5 mM EGTA, pH9.0) and subjected to heat-induced epitope retrieval for 20 min before blocking. All antibodies were diluted in DPBS + 0.3%BSA + 0.3% TritonX-100. Paraffin-embedded sections were analyzed using a confocal fluorescence microscope (NIKON A1

Resonant Confocal Microscope) and images were captured using the Nikon Elements software. Antibodies are listed in Supplementary Tables 4 and 5.

**Scanning electron microscopy**. After standard scanning electron microscopy (SEM) tissue preparation, the surface morphologies of the hPSC-derived cholangiocytes were sputter coated with gold-palladium prior to loading in the SEM. Images were observed using a SU3500 scanning electron microscope (Hitachi, Japan) with a 5 kV magnification.

**Quantitative real-time PCR**. Total RNA was prepared using RNAqueous Micro Kit (Invitrogen) and treated with RNase-free (Ambion). In all, 500 ng to 1 µg RNA was reverse-transcribed into cDNA using iSCRIPT Reverse Transcription Supermix (BIO RAD). qPCR was performed on a C1000 Touch Thermal Cycler (BIO RAD) using a SsoAdvanced Universal SYBR Green Supermix (BIO RAD). Expression levels were normalized to the housekeeping gene TATA box binding protein (TBP) as previously described[21]. Oligonucleotide sequences are available in Supplementary Table 6. Control samples of RNA of adult liver (AL), fetal liver (FL), gall bladder (GB), and pancreas (PANC) are listed in Supplementary Table 7.

**Apical chloride conductance assay**. Apical CFTR-mediated chloride conductance was measured as previously described[18]. Briefly, hPSC-derived cholangiocytes were grown in 96-well plates and treated for 24 h with either or combination of CFTR modulators, 3 µM VX809, 3 µM VX661 (Selleck), 3 µM R and S-VX445 (MedChemExpress), 0.5 µM AC1 (X281605), 3 µM AC2-1 (X281632), 3 µM AC2-2 (X300549) (Abbvie), or DMSO. Cells were labeled using blue membrane potential sensitive FLIPR dye dissolved in sodium and chloride-free buffer (150 mM NMDG, 150 mM gluconolactone, 10 mM HEPES, pH 7.4, 300 mOsm) at a concentration of 0.5 mg/ml. Cells were incubated for 30 min at 37 °C. The plate was read in a fluorescence plate reader (FLIPR® Tetra System or SpectraMax i3; Molecular Devices) at 37 °C. After reading baseline fluorescence, CFTR was stimulated with a combination of the cAMP agonist Forskolin (10 µM) and potentiators, 1 µM VX770 (Selleck) or 1.5 µM AP2 (X300529) (Abbvie). CFTR-mediated depolarization was detected as an increase in fluorescence, and repolarization was detected as a decrease with addition of 10 µM CFTR-specific inhibitor CFTR$_{Inh}$-172 to all wells.

**Western blotting**. Cells were lysed in modified radioimmunoprecipitation assay buffer (50 mM Tris-HCl, 150 mM NaCl, 1 mM EDTA, pH 7.4, 0.1% SDS, and 0.1% Triton X-100) containing a protease inhibitor cocktail (Roche) for 10 min, and the soluble fractions were analyzed by sodium dodecyl sulfate-polyacrylamide gel electrophoresis (SDS-PAGE) on 6% Tris-Glycine gels (Life Technologies, Carlsbad, CA)[68]. After electrophoresis, proteins were transferred to nitrocellulose membranes and incubated in 5% milk. CFTR bands were detected with human CFTR IgG2b mAb596 (1:2000, University of North Caroline, Chapel Hill, NC, code: A4, Cystic Fibrosis Foundation Therapeutics Inc.) overnight at 4 °C and horseradish peroxidase-conjugated goat anti-mouse IgG secondary antibody (1:5000, Pierce, Rockford, IL) for 1 h at RT. Loading control proteins (actin or calnexin) bands were detected with anti-α-actin mouse pAb (1:5000, Sigma, St. Louis, MO, cat. no.: A5044) or calnexin-specific rabbit pAb (1:5000, Sigma, St. Louis, MO, cat. no.: C4731) primary antibody overnight at 4°C and horseradish peroxidase–conjugated goat anti-rabbit IgG secondary antibody (1:5000, Pierce) for 1 h at RT. Wt CFTR and calnexin antibodies were exposed using Amersham ECL Western Blotting Detection Reagent (GE Healthcare Life Sciences, Canada) for 1–5 min exposures on the Li-Cor Odyssey Fc (LI-COR Biosciences, Lincoln, NE). Relative expression level of CFTR proteins were quantified by densitometry of immunoblots, using Image Studio Lite software (Version 5.2.5) from Li-Cor Odyssey Fc (LI-COR Biosciences)[69]. Uncropped blots are available in Source Data and Supplementary Information.

**CFTR swelling assay**. Day 6–12 3D cholangiocytes were replaced in 1 well of 384 well plate with Hank's buffered solution and imaged using live cell imaging at 37 °C (ZEISS Axio Observer). cAMP agonists- Forskolin (10 µM) or secretin (100 µM, Sigma) were added and images were taken every 15 min for 1 h. One well in each experiment was pre-incubated with 50 µM CFTR-specific inhibitor-Inh172 (Selleck Chemicals) or DMSO control for 1 h. CF-patients and mutation corrected CF iPSC-derived cholangiocytes were treated with either 3 µM VX809 (Selleck), or 0.5 µM AC1 and 3 µM AC2-2 (Abbvie) for 24 h before the assay. CFTR was stimulated with the cAMP agonist FSK (10µM) and potentiators, 1 µM VX770 (Selleck) or 1 µM AP2 (Abbvie). Data were captured using Zeiss Zen software (2.3). The total area of each cyst was calculated before and 1 h after stimulation using ImageJ (NIH-version 1.51). The area of each cyst after 1 h stimulation was calculated and plotted.

**Single-cell RNA-seq data processing and analysis**. Cholangiocytes were generated from H9 ES line, dissociated to single cells at day 55 of 2D and 3D cholangiocytes. 6313 cells (2D) and 5892 cells (3D) were sequenced with the depth of 70,000 reads per cell. Samples were processed using 10x Genomics Single Cell 3′ v3.1 Reagent Kit. scRNA-seq data associated with hPSC-derived cholangiocytes are

deposited in GEO (GSE175502). For raw data processing, Chromium Single Cell Software Suite (v3.1) was used for sample demultiplexing, read alignment, barcode processing, and UMI counting. Cellranger mkfastq was then used to generate FASTQ files from BCL files. Subsequently, cellranger count was used to generate single-cell gene counts. Reads in the FASTQ files were mapped to the human reference genome (NCBI build38/UCSC hg38) with STAR software[70]. Of note, only the exonic loci were prioritized and they were confidently mapped to the exonic loci with MAPQ 255. Chromium cellular barcodes were used to generate gene-barcode matrices. Only reads that were confidently mapped to the transcriptome were used for the UMI count. Filtered gene-barcode matrices containing only cellular barcodes in the MEX format were used for downstream analyses. Analyses of the processed count matrix were performed with previously published R packages (R v3.6.1) mainly including Seurat (v3.2.2), batchelor (v1.2.4) and clusterprofiler (v3.14.3)[71–74]. The filtered gene-barcode matrices of the 2D and 3D datasets from the cell ranger pipeline were fed into the Seurat package and merged as a single object for downstream analyses. First, the raw read counts generated by cellranger were filtered based on mitochondrial contents, feature number and RNA number (nCount_RNA > 800 & nFeature_RNA > 2000 & percent.mt <25), and then they were normalized by a global-scaling normalization method "NormalizeData" which normalizes the gene expression measurements for each cell by the total read counts. The top 2000 most variably expressed genes were identified (FindVariableFeatures) and used as the features of the dataset. Given the inherited batch effects in the 2D and 3D datasets, the log normalized counts of the two datasets were corrected using the method mutual nearest neighbor (mnn) with RunFastMNN function originally published as a part of the batchelor package (v1.2.4) and adopted by the Seurat-wrapper package (v0.3.0). The corrected data were used for Uniform Manifold Approximation and Projection (UMAP) (RunUMAP function; reduction = "mnn", dims = 1:30). For clustering, resolutions of 0.7 (FindClusters function) was used for the data shown in the present study. To identify marker genes or upregulated genes, the clusters of interest were subset and compared for differential gene expression using the Wilcoxon rank-sum test (FindAllMarkers function; only.pos = TRUE, min.pct = 0.25, logfc.threshold = 0.15, p-value cut-off = 0.05). Gene ontology analysis was performed using the clusterprofiler package. Genes upregulated in the selected clusters were obtained using the FindAllMarkers function, and the genes enriched in these clusters were subsequently analyzed for the enrichment of biological processes (BP) using the compareCluster function (function = enrichGO, ontology = BP, pvalueCutoff = 0.05). The human liver dataset was from the study by MacParland et al. (GSE115469)[50]. The cells were filtered based on the feature number (nFeature_RNA > 1500) and mitochondrial content (percent.mt <50) described in the manuscript. The annotation of the four main cell types that include cholangiocyte, hepatocyte, immune cell, and endothelial cell was based on the markers published by MacParland et al.[50]. The integration of 2D, 3D, and the human liver datasets was achieved by using the mnn pipeline described above. To calculate the Pearson correlation between the cells generated in vivo and in vitro, the expression levels of the 2000 variable features was extracted from the 'RNA' slot and subsequently used for the calculation of Pearson correlation coefficient. Finally, the correlation matrix was visualized using the corrplot package (v 0.84). Sample numbers analyzed in this study were: monolayer hPSC-derived cholangiocyte ($n = 1$), cyst hPSC-derived cholangiocyte ($n = 1$).

**Calcium signaling assay**. GCaMP 3 (gift from Michael Laflamme Lab, Toronto)[44,47] and H9, CF01 iPSC were differentiated to cholangiocyte cysts and plated down in Lab-Tek Chamber Slide (Millipore) or in flow chamber (ibidi) 2 days prior to the assay. Day 27 hepatoblasts were aggregated and plated down 2 days prior to the assay. Chambers were pre-coated with fibronectin. The cells were replaced in Tyrode buffer before the assay. H9-derived-cholangiocytes and CF01 derived cholangiocytes were stained with 10 µM Fluo4 (Invitrogen) with 0.04% Pluronic F127 (Invitrogen) for 30 min prior to the assay. To measure calcium response to ATP and TUDCA (Sigma) in 3D cysts, GCaMP derived cholangiocyte 3D cysts were embedded in the type1 collagen gel just before the assay.

Calcium response to 20 µM ATP (Sigma) or flow (Perista BioMini Pump: ATTO) were analyzed using a confocal fluorescence microscope (NIKON A1 Resonant Confocal Microscope) and images captured using the Nikon Elements software.

**ACC assay under the flow**. Cholangiocytes were prepared and plated down the same way as described in calcium signaling assay. CF patient-derived cholangiocyte were treated as the same concentration of CFTR modulators and stained with FLIPR solution as mentioned in ACC assay. In all, 2 µM Thapsigargin (Sigma) was used for calcium inhibition. Images were acquired using Leica SP8/STED confocal microscope. Data were processed with Velocity software suite (PerkinElmer).

**Mice and transplantation**. All animal experiments were carried out under procedural guidelines, severity protocols and with ethical approval from the Central Institute for Experimental Animals (CIEA) in Japan, and University Health Network Animal Care Committee (Toronto). All mice were housed under 12 light/ 12 dark cycle, temperatures of 22 ± 2 °C with 50 ± 10% humidity. TK-NOG mice transplanted at the Central Institute for Experimental Animals (CIEA) in Japan. Ganciclovir (GCV: 10 mg/kg) was administered into 7–8 weeks-old adult male TK-

NOG mice 7–10 days prior to the transplantation. The details of induction of liver injury and transplantation were mentioned previously[51,75]. Briefly, one week after GCV treatment, the degree of liver damage was examined by determining plasma alanine aminotransferase (ALT) levels using an automated clinical chemistry analyzer (FUJI DRI-CHEM 7000; Fuji Photo Film, Tokyo, Japan). Dissociated cells were intrasplenically injected using a ½ mL Insulin Syringe with Permanently Attached Needle (29 G x ½-in+ Termo, Tokyo, Japan). NSG male mice purchased from The Jackson Laboratory (Bar Harbor, ME) were kept in a specific pathogen-free mouse facility at the Princess Margaret Cancer Research Tower (PMCRT) and were used at 8–10 weeks of age. Day 55 monolayer cholangiocytes were dissociated by trypsin-EDTA and one million cells were injected into the spleen of TK-NOG mice (CIEA) or the kidney subcapsular of NSG mice (PMCRT). At the indicated time points following after the transplantation, animals were euthanized, and their tissue was isolated and fixed with 10% formalin. Fixed samples were embedded in paraffin, sectioned, and stained with hematoxylin and eosin (H and E) for morphological analyses.

**Statistics and reproducibility**. Statistical methods and the numbers of biological replicates were indicated in figure legends. All experiments were repeated a minimum of three times. SEM was calculated using data from biological and technical replicates. $p < 0.05$ was considered statistically significant (* or # <0.05, ** or ## <0.01, *** or ### <0.001, **** or #### <0.0001). Statistical analysis was performed using GraphPad Prism 8.

**Reporting summary**. Further information on research design is available in the Nature Research Reporting Summary linked to this article.

## Data availability

The data supporting the findings of this study are available within the Article and its Supplementary Information files, and from the corresponding author on reasonable request. A Reporting Summary for this Article is available as a Supplementary Information file. Source data are provided with this paper. Raw scRNA seq data generated in this study has been deposited at the GEO database under accession code: GSE175502, referenced scRNA seq data of human adult liver is available under accession code: GSE115469. Source data are provided with this paper.

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

## Acknowledgements

We thank Dr. Michael Laflamme for providing GCaMP cell line, and Abbvie for providing the reference tool compounds AC1, AC2-1, AC2-2, and AP2. Dr. Sonya Mac-Parland, Dr. Gary Bader, and Dr. Ian McGilvray for providing the Counts Matrix data of primary human adult liver. We also thank Dr. Gordon Keller for his invaluable guidance and assistance. Induced pluripotent stem cells were obtained through the CF Canada-SickKids Program for Individualized CF Therapy (CFIT). The CFIT Program is jointly funded by the Sick Kids Foundation and Cystic Fibrosis Canada.This work is supported by the CFIT Program (C.E.B.), and JSPS KAKENHI grant number JP18K08589 (S.O.). This research is part of the Government of Canada through Genome Canada and the Ontario Genomics Institute (OGI-148) (awarded to C.E.B. and S.O.), and part of the University of Toronto's Medicine by Design initiative, funding from the Canada First Research Excellence fund (awarded to C.E.B. and S.O.).

## Author contributions

M.O. and S.O. performed experimental design, data acquisition, analysis, interpretation, and wrote the paper. J.J., S.X. and D.Y. performed the CFTR functional experiments and data analysis. D.Y. performed scRNA seq analysis. J.J., S.X. and D.Y. equally contributed to this study. A.D., M.H., C.C., O.L. and Y.H. performed experiments. C.D. provided technical support. H.S. and M.G. provided experimental guidance. C.E.B. and S.O. supervised experiments, edited, revised, and performed the final approval of the paper.

## Competing interests

OHSU has commercially licensed HPd3/DHIC5-4D9; authors C.D. and M.G. are inventors of this antibody. The remaining authors declare no competing interests.
