## [Peer Review File · Nature Communications]

REVIEWER COMMENTS

Reviewer #1 (Remarks to the Author):

In this interesting study, the authors generate and describe a new method for generating human cholangiocytes in vitro. The authors demonstrate preliminary findings that these human pluripotent stem cell (hPSC)-derived cholangiocytes have some markers of mature cholangiocytes, and can function in both 2D and 3D cultures. The 3D hPSC-cholangiocytes are responsive to CFTR inhibitor treatments, indicating that this method could be used for drug screening. Furthermore, these hPSC-cholangiocytes were injected into mouse models and were able to generate bile ducts. This manuscript is very intriguing and brings a new approach to the field. Additionally, the in vivo studies are novel and new for the field. However, this reviewer believes that looking into cholangiocyte maturation and heterogeneity is further needed, as this is an area that is usually not discussed in these induced cholangiocyte models and would greatly enhance the quality of the paper. As well, it is unclear what the benefits and uses of the 3D versus 2D models with these cells, as both models seem useful and viable. In general, this is an exciting and interesting paper, but further work on clarifying the maturation and function of these cholangiocytes is needed.

Major comments:

1. It is important that the authors also look at other markers of mature cholangiocytes; such as, secretin, secretin receptor and AE2. While CFTR and the presence of cilia are important, mature cholangiocyte markers these other factors may be key as well, considering loss of AE2 is associated with cholangiopathies, such as PBC. This is particularly the case for Figure 1, and some for Figure 2 (though it contains secretin receptor gene expression).
2. The data in Figure 2 showing maturation of hPSCs to cholangiocytes is a bit subjective considering it is primarily based on qPCR data. The paper claims that the flow cytometry analyses confirmed the molecular analyses; however, the supplemental figure only shows flow data for CK7 and EPCAM (which are inherent biliary markers and not indicative of maturation), and AFP and ALB (which is to prove no contamination of hepatocytes. Therefore, this data should have been used to mark mature as well as immature cholangiocyte markers to see what kind of population had arisen from the hPSC. Similarly for the staining, some markers (ASBT and alpha-tubulin) are suited for mature cholangiocyte markers, but others (secretin, SR, CFTR, AE2) should have been included as well.
3. Mature cholangiocytes would also respond to secretin treatment and would demonstrate increased cAMP and calcium levels, chloride efflux (if CFTR is functionally active), bicarbonate release (assuming the CFTR/AE2 axis is functional) and thus changes in pH. The authors should have evaluated these parameters to ensure cholangiocyte maturity. Furthermore, this reviewer is wondering about aquaporin expression and water transport?
4. More conceptually, do the authors believe they have a virtually pure population of mature/large cholangiocytes, or a mixture of both small and large cholangiocyte phenotypes? It is known that small cholangiocytes do not express CFTR, which seems to be abundant in the monolayer. It would be interesting if this could be clarified as it would help distinguish applicability for different cholangiopathies.
5. Do the cells in monolayer express polarity?
6. In general, what is the cellular phenotype of the hPSC-derived cholangiocytes? In general, normal cholangiocytes are mitotically dormant, but can become proliferative, inflammatory/reactive, neuroendocrine or senescent in response to injury. This reviewer is curious if these cholangiocytes are normal until being injured, or if they inherently express proliferation, etc. markers.
7. The experiments evaluating CFTR inhibitors are quite intriguing, but it would be confirming if the authors evaluated chloride efflux and bicarbonate release that are mediated by CFTR opening.
8. This reviewer is confused why the authors only evaluated calcium signaling in the 3D cultures but not the 2D? Based on gene expression, which may not be the best indicator, it seems as if the mature

cholangiocyte markers are decreased in the 3D organoids when compared to the 2D culture, can the authors explain this discrepancy? For this reason, it isn't clear why the authors generated a 3D culture and what are the key benefits of using this system versus the 2D monolayer?

9. Are the bile ducts in the mice that are generated from the hPSC-cholangiocytes functional?

10. Do the cilia of the hPSC-cholangiocytes express TGR5?

11. Are the hPSC-cholangiocytes responsive to bile acids?

Minor comments

1. The monolayer seems quite thick and dense? This reviewer is wondering if the cell layer is too confluent and if this would have any effect on the cells?

2. Does the injection of the hPSC-derived cholangiocytes increase any damaging effects, such as inflammation or fibrosis?

Reviewer #2 (Remarks to the Author):

In the submitted manuscript, the authors described a new strategy of generating mature functional cholangiocytes from human pluripotent stem cells (hPSCs), and the potential application of generated cholangiocytes for the bio-medical studies of cholangiopathies, a group of liver diseases with different etiology but with the same target cells, i.e., cholangiocytes. In this study, the authors generated cholangiocytes from hPSCs of healthy humans and from patients with mutations in gene encoding CFTR (the Cystic Fibrosis Transmembrane conductance Regulator) to show in particular that these cell lines are applicable for testing the efficacy of new drugs against cystic fibrosis. The authors work is significant because the current needs in cholangiocytes from patients with different cholangiopathies are essential. The manuscript clearly shows that the authors achieved their goal and generated cholangiocytes from hPSCs of healthy humans and patients with cystic fibrosis. It is a well-conceived, executed, illustrated and described work. However, some interpretations of functional studies and statements regarding primary cilia are not convincing and require attention.

Major comments:

First, the authors stated that only mature cholangiocytes possess primary cilia, i.e., the existence of cilia indicates the maturity of cholangiocytes (page 6). This notion is overstated. It is known, for example, that human embryonic stem cells have primary cilia that play an important role in embryonic development and tissue differentiation (see for example Kiprilov et al., JCB 2008; 180: 897-904). Thus, the authors should provide additional information regarding primary cilia in hPSCs from which they generated mature cholangiocytes.

Second, an increase in expression of TRPV4, PKD1 and PKD2 (page 6, Figure 2d) does not directly reflect changes in ciliogenesis because the proteins encoded by these genes have multiple intracellular localization and functions.

Third, the formation of primary cilia is strongly linked to the cell cycle. Because the authors used a number of agonists and antagonists of the pathways crucial for bile duct development and maturation (page 6), they have to experimentally address if these drugs affect the cell cycle (multiple experimental approaches to assess the cell cycle are available).

Fourth, it is not clear why the authors used a model of 3D cysts/organoids from generated cholangiocytes to study ciliary-mediated intracellular Ca²⁺ signaling in response to ATP and fluid flow applied to the basolateral plasma membrane (page 11). The description of experiments with "the cyst-

derived cholangiocytes" and "open cysts" (i.e., they become open) exposed to the fluid flow requires clarification. Also, the images of cholangiocyte cilia shown in Figure 7a are not convincing and should be replaced by better images. There is no evidence provided that cholangiocyte cilia are involved in ATP and fluid-flow mediated intracellular Ca²⁺ signaling. To support this conclusion, additional experiments with inhibited ciliogenesis in cholangiocytes are needed.

Fifth, the mechanistic part of the rescue of the CFTR function after the treatment with a drug against cystic fibrosis (VX809) combined with an increased fluid flow (page 12) is missing.

Sixth, the authors' conclusion on the role of primary cilia in regulation of CFTR in generated cholangiocytes via the cilia-mediated intracellular Ca²⁺ signaling pathway (page 15) is premature and requires additional studies on cholangiocytes in which formation of cilia is prevented.

Minor comments:

- In Figure 1a, "Cholangiocyte Day 37" and "Ciliated Cholangiocyte" should be plural – "Cholangiocytes".
- Supplementary Video 1 shows primary cilia in H9(?) cholangiocytes. H9 is confusing, because in the text (page 7) the authors are talking about DHIC5-4D9+. Are DHIC5-4D9+ cholangiocytes H9 cholangiocytes? Needs clarification.
- In Figure 3b, the data on "d49 RA" are missing.
- In Figure 5d,e instead of "DMSO" should be "FSK".
- On page 10 and in Figure 5d, the authors presented data generated on H9 hESC-derived (?) cholangiocyte cysts. What are H9 hESC-derived cholangiocyte cysts? Needs clarification.
- The authors need to provide a list of abbreviation that will help the readers to follow the text.

Reviewer #3 (Remarks to the Author):

In the submitted manuscript the authors describe the production of cholangiocytes from human iPSCs that display an array of functional activities that reproduce key aspects of bile duct function. They go on to show that the cells respond to drugs for the treatment of cystic fibrosis and could form a platform for drug discovery. The study is well conceived and the data are of high quality. Although others have reported the production of bile duct cells from iPSCs previously, this represents a substantial advance in both efficiency and maturity of the cells that are generated. Most importantly the cells produce cilia that have key roles and are known to contribute to cholangiopathies. While enthusiasm for the project is high there are specific areas that could be addressed to further improve the manuscript:

It would be helpful to have a more global profiling of expression to compare to endogenous human cholangiocytes. While the RT-qPCR is convincing, RNA-seq analyses would be an unbiased approach that would provide a greater understanding of the state of the iPSC-cholangiocytes and whether there are changes that could affect function.

In figure 4 the authors distinguish responses to CFTR drug treatments by iPSC-cholangiocytes from different patients. The patients share the same CFTR variation yet they exhibit distinct responses to the drug treatments. This is provocative and has the potential to be of value when defining the course of treatments for specific patients. However, there is some concern that the variations do not reflect the patient responses but instead are underlying differences between clonal iPSC lines. It would be useful to examine the response in multiple distinct iPSC clonal cell lines derived from a single

individual's (isogenic) somatic cells with a 'normal' CFTR allele. One would predict that such iPSC-derived cholangiocytes have a relatively similar response between clones.

Minor points:

Some of the labelling of the bar charts was a bit difficult to follow and required a bit of digging to work out what the data represented. Labelling could be clarified throughout the manuscript.

In Figure 1B it would be better to use a different color scheme on the bar graphs because red/green can be problematic for those with color-blindness.

In Figure 1 C.D., it wasn't intuitive how the authors knew that the CFTR bands were glycosylated with glycosylase treating the extracts. Some clarification could be helpful.

In Figure 4, are the microtiter wells referred to by the authors equivalent to the 96-well plates used to determine the z-factor.

In Figure 5 the swelling response comparing mutant to wild type cells is convincing; however, it wasn't clear whether the relatively subtle changes would be quantitative enough to differentiate between potential treatment options. It could be helpful to show a dose response curve relating swelling to drug treatment.

The reviewers of the original submission raised a number of valid concerns and made good suggestions for additional experiments that proved helpful in reshaping our revised manuscript. We have addressed all of the reviewers' comments that in many cases required additional experiments. We have focused on addressing the specific reviewers' questions, (1) we performed single cell RNAseq analysis to assess further markers/hall marks of maturation, and clarify the benefits/difference between 2D and 3D culture and to assess the cell cycle status in our differentiated cholangiocytes, (2) we analyzed the CFTR functional activity assay of CF iPS derived cholangiocytes using two independent sub-clonal iPS cells along with CFTR gene corrected lines to assess clonal responses to drugs, (3) we performed the fluid response assay on non-ciliated cholangiocytes to investigate the functional impact of primary cilia and (4) we further characterized the *in vivo* engraftment and confirmed that human hPSCs derived cholangiocytes form luminal ductal structures with existing mice bile duct epithelial cells.

Based on these findings, we are able to provide the first evidence that (1) CF patients' iPS derived cholangiocytes can be used to predict therapeutic efficacy, (2) primary cilia related physiological function in hPSC derived cholangiocytes can be measured under the fluidic condition establishing a powerful new platform to study primary cilia diseases, (3) either 2D or 3D formats can be adaptable to model/study biliary diseases, and (4) it is possible to generate new biliary ductal structures by cholangiocyte transplantation, paving the way for developing new cell based therapies to treat cholangiopathies.

Please find, a point-by-point response (colour highlighted) to each of the reviewer's comments. Changes are also shown in yellow in the revised paper.

REVIEWER COMMENTS

Reviewer #1 (Remarks to the Author):

In this interesting study, the authors generate and describe a new method for generating human cholangiocytes *in vitro*. The authors demonstrate preliminary findings that these human pluripotent stem cell (hPSC)-derived cholangiocytes have some markers of mature cholangiocytes, and can function in both 2D and 3D cultures. The 3D hPSC-cholangiocytes are responsive to CFTR inhibitor treatments, indicating that this method could be used for drug screening. Furthermore, these hPSC-cholangiocytes were injected into mouse models and were able to generate bile ducts. This manuscript is very intriguing and brings a new approach to the field. Additionally, the *in vivo* studies are novel and new for the field. However, this reviewer believes that looking into cholangiocyte maturation and heterogeneity is further needed, as this is an area that is usually not discussed in these induced cholangiocyte models and would greatly enhance the quality of the paper. As well, it is unclear what the benefits and uses of the 3D versus 2D models with these cells, as both models seems useful and viable. In general, this is an exciting and interesting paper, but further work on clarifying the maturation and function of these cholangiocytes is needed.

Major comments:

1. It is important that the authors also look at other markers of mature cholangiocytes; such as, secretin, secretin receptor and AE2. While CFTR and the presence of cilia are important, mature cholangiocyte marker these other factors may be key as well, considering loss of AE2 is associated with cholangiopathies, such as PBC. This is particularly the case for Figure 1, and some for Figure 2 (though it contains secretin receptor gene expression).

We agree that the expression of these genes is important and had included analyses of SLC4A2 (AE2), secretin receptor (SCTR) in Figure 2d of our original manuscript. The day37 population in Figure 2d is the same stage population used Figure 1b for screening to identify the signaling pathways that upregulate CFTR. We observed that both AE2 and SCTR were upregulated at day37 and day49 of differentiation.

2. The data in Figure 2 showing maturation of hPSCs to cholangiocytes is a bit subjective considering it is primarily based on qPCR data. The paper claims that the flow cytometry analyses confirmed the molecular analyses; however, the supplemental figure only shows flow data for CK7 and EPCAM (which are inherent biliary markers and not indicative of maturation), and AFP and ALB (which is to prove no contamination of hepatocytes). Therefore, this data should have been used to mark mature as well as immature cholangiocyte markers to see what kind of population had arisen from the hPSC. Similarly for the staining, some markers (ASBT and alpha-tubulin) are suited for mature cholangiocyte markers, but others (secretin, SR, CFTR, AE2) should have been included as well.

We agree with reviewer's comments and have included immunostaining analyses of secretin receptor (SCTR), and SLC4A2 (AE2). CFTR was included in Figure 2e of our original manuscript. We included these images in the new Figure 2e and Supplementary Figure 2a. These data are shown in Figure 2e and Supplementary Figure 2a.

3. Mature cholangiocytes would also respond to secretin treatment and would demonstrate increased cAMP and calcium levels, chloride efflux (if CFTR is functionally active), bicarbonate release (assuming the CFTR/AE2 axis is functional) and thus changes in pH. The authors should have evaluated these parameters to ensure cholangiocyte maturity. Furthermore, this reviewer is wondering about aquaporin expression and water transport?

We conducted additional studies to test for the functional expression of the secretin receptor and secretin mediated signaling as suggested. We found that the addition of secretin induced cAMP-mediated, CFTR dependent swelling of cholangiocyte spheroids. These new data are shown in figure 5d and supplementary video 3. We have detected AQP1 expression by scRNA seq (Figure 8c and f) and qPCR analyses (Supplementary Figure 13). The levels are considerably higher in the 3D cholangiocyte cysts than in the cells generated in monolayer cultures. We agreed that bicarbonate release and variation of pH would be important aspects of function in mature cholangiocytes. We are now developing an assay to interrogate the functional interaction between CFTR and AE2 in cholangiocyte cysts. We plan to publish findings from these studies and others regarding differential transporter interactions in non-CF and CF cysts in a future manuscript.

4. More conceptually, do the authors believe they have a virtually pure population of mature/large cholangiocytes, or a mixture of both small and large cholangiocyte phenotypes? It is known that small cholangiocytes do not express CFTR, which seems to be abundant in the monolayer. It would be interesting if this could be clarified as it would help distinguish applicability for different cholangiopathies.

To address this question, we conducted scRNA seq analyses to investigate the heterogeneity in hPSC-derived cholangiocyte populations generated in both monolayers and 3D cysts. As shown in Figure 8a, we observed subclusters in the cholangiocyte population although almost all clusters express pan-cholangiocyte markers including CFTR. Comparison of the molecular profiles of our hPSC-derived populations to published data basis suggests that we have generated cholangiocytes that form the intrahepatic bile duct. Given this, we believe our cells are appropriate for modeling the majority of the cholangiopathies. Our ability to further characterize our cells is currently limited by the lack of detailed transcriptional analyses of primary human cholangiocytes. As these data become available, we will be able to further identify the subpopulations of cholangiocytes that we have generated.

5. Do the cells in monolayer express polarity?

As shown in Supplementary Figure 2b, 2c, and 2d, primary cilia, CFTR and ZO-1 proteins are expressed

on the apical side of cells indicating polarization of the cells in vitro.

6. In general, what is the cellular phenotype of the hPSC-derived cholangiocytes? In general, normal cholangiocytes are mitotically dormant, but can become proliferative, inflammatory/reactive, neuroendocrine or senescent in response to injury. This reviewer is curious if these cholangiocytes are normal until being injured, or if they inherently express proliferation, etc. markers.

Analyses of our scRNA seq data revealed the presence of a small cluster of cells in the G2/M phase cell cycle that expressed CDK1, BRCA2 and CDC6, genes involved in cell proliferation (Figure 8d and 8e). These cells expressed cholangiocyte markers. These data are described on page 15 line 4.

7. The experiments evaluating CFTR inhibitors are quite intriguing, but it would be confirming if the authors evaluated chloride efflux and bicarbonate release that are mediated by CFTR opening.

We measured CFTR, chloride channel activity in the cholangiocytes using two assays. In the first, we used forskolin to induce cyst swelling as it is known to activate CFTR. Second, we measured CFTR channel function as changes in apical membrane potential. This channel activity assay has been well documented in the literature (Ahmadi 2017 doi: 10.1038/s41525-017-0015-6). As mentioned above, we plan to report on the functional interaction between CFTR and other transporters in our future work.

8. This reviewer is confused why the authors only evaluated calcium signaling in the 3D cultures but not the 2D? Based on gene expression, which may not be the best indicator, it seems as if the mature cholangiocyte markers are decreased in the 3D organoids when compared to the 2D culture, can the authors explain this discrepancy? For this reason, it isn't clear why the authors generated a 3D culture and what are the key benefits of using this system versus the 2D monolayer?

We clarify the rationale for generating cholangiocytes in both monolayer cultures and 3D cysts in the revised manuscript, page 11 from line 16. Basically, the 2D format is useful for the direct measurement of CFTR and robust high throughput screening whereas the 3D cysts are well suited for indirect functional CFTR measurement, as the receptor is expressed inside of the lumen. We observed higher expression levels of AQP1 in 3D cysts (Figure 8c, 8f and Supplementary Figure 13) indicating that these cells may display a more physiological response under certain conditions. The cysts are also useful for seeding the ibidi flow chambers for measuring the calcium signaling under the fluid flow as dissociated monolayer cells lose their primary cilia and do not survive well under these conditions. Additionally, scRNA seq analysis revealed differential gene expressions in 2D format and 3D cysts (Figure 8h and Supplementary Figure 15c). The data are described on page 15 line 13.

9. Are the bile ducts in the mice that are generated from the hPSC-cholangiocytes functional?

We stained the engrafted livers with both human and mouse specific anti-CK19 antibodies and found that the cells co-exist in proximity, suggesting that they may be part of the same structure. These data are shown in Figure 9c and described in the revised manuscript page 16 line 8. Formal demonstration that the human cells in these livers contribute to functional bile ducts will require additional studies that are beyond the scope of this study.

10. Do the cilia of the hPSC-cholangiocytes express TGR5?

We stained the cells with an anti-TGR5 antibody (Invitrogen PA5-27076) but were unable to detect any expression on the cilia. Data is shown in the Supplementary Figure 2b.

11. Are the hPSC-cholangiocytes responsive to bile acids?

We analyzed responsiveness to bile acid and found that TUDCA activates calcium signaling in hPSC-derived cholangiocytes. These data are shown in Figure 6d and Supplementary Video 5 and described in the revised manuscript on page 13 line 7.

Minor comments

1. The monolayer seems quite thick and dense? This reviewer is wondering if the cell layer is too confluent and if this would have any effect on the cells?

Based on our immunohistochemistry analyses, we estimate the monolayer to be approximately 10 to 15µm thick. The cells express tight junction marker ZO-1 as shown in Figure 3a, Supplementary Figure 2a and 2d.

2. Does the injection of the hPSC-derived cholangiocytes increase any damaging effects, such as inflammation or fibrosis?

The cholangiocytes are injected into the spleen from where they migrate through the splenic-portal vein plexus to the liver parenchyma. This does not damage the liver (Higuchi 2016 doi.org/10.1111/hepr.12644).

Reviewer #2 (Remarks to the Author):

In the submitted manuscript, the authors described a new strategy of generating mature functional cholangiocytes from human pluripotent stem cells (hPSCs), and the potential application of generated cholangiocytes for the bio-medical studies of cholangiopathies, a group of liver diseases with different etiology but with the same target cells, i.e., cholangiocytes. In this study, the authors generated cholangiocytes from hPSCs of healthy humans and from patients with mutations in gene encoding CFTR (the Cystic Fibrosis Transmembrane conductance Regulator) to show in particular that these cell lines are applicable for testing the efficacy of new drugs against cystic fibrosis. The authors work is significant because the current needs in cholangiocytes from patients with different cholangiopathies are essential. The manuscript clearly shows that the authors achieved their goal and generated cholangiocytes from hPSCs of healthy humans and patients with cystic fibrosis. It is a well-conceived, executed, illustrated and described work. However, some interpretations of functional studies and statements regarding primary cilia are not convincing and require attention.

Major comments:

First, the authors stated that only mature cholangiocytes possess primary cilia, i.e., the existence of cilia indicates the maturity of cholangiocytes (page 6). This notion is overstated. It is known, for example, that human embryonic stem cells have primary cilia that play an important role in embryonic development and tissue differentiation (see for example Kiprilov et al., JCB 2008; 180: 897-904). Thus, the authors should provide additional information regarding primary cilia in hPSCs from which they generated mature cholangiocytes.

We agree with that primary cilium are expressed in the different cell types. However, within the liver, only the cholangiocytes express these cilia. As shown in different rodent studies, the cilia on the cholangiocytes playing a critical role in regulating bile duct fluid secretion (LaRusso 2011 doi: 10.1159/000324121, Masyuk 2006 doi: 10.1053/j.gastro.2006.07.003, Masyuk 2008 doi: 10.1152/ajpgi.90265.2008). The ability to generate ciliated cholangiocytes from hPSCs enables us to investigate the role of these structure in the human cells.

Second, an increase in expression of TRPV4, PKD1 and PKD2 (page 6, Figure 2d) does not directly

reflect changes in ciliogenesis because the proteins encoded by these genes have multiple intracellular localization and functions.

We agree with the reviewer and have analyzed the expression patterns of additional ciliary genes in our scRNA seq data set (Supplementary Figure 15b)

Third, the formation of primary cilia is strongly linked to the cell cycle. Because the authors used a number of agonists and antagonists of the pathways crucial for bile duct development and maturation (page 6), they have to experimentally address if these drugs affect the cell cycle (multiple experimental approaches to assess the cell cycle are available).

We agree that is an important question and we examined the cell cycle status of the population in our scRNA seq analyses. As shown in Figure 8c and 8d, cluster 7 expresses CDK1, BRCA2, CDC6 and these cells are in G2/M cycle. We described these findings in the revised manuscript, on page 15 line 4.

Fourth, it is not clear why the authors used a model of 3D cysts/organoids from generated cholangiocytes to study ciliary-mediated intracellular Ca²⁺ signaling in response to ATP and fluid flow applied to the basolateral plasma membrane (page 11). The description of experiments with “the cyst-derived cholangiocytes” and “open cysts” (i.e., they become open) exposed to the fluid flow requires clarification. Also, the images of cholangiocyte cilia shown in Figure 7a are not convincing and should be replaced by better images. There is no evidence provided that cholangiocyte cilia are involved in ATP and fluid-flow mediated intracellular Ca²⁺ signaling. To support this conclusion, additional experiments with inhibited ciliogenesis in cholangiocytes are needed.

This is a very important issue that we have clarified in the revised manuscript. 3D cysts were plated in the flow chamber (ibidi) without any dissociation, as we observed that dissociation disrupts cilia expression. The cysts adhere to the bottom of the chamber and open, exposing the cholangiocytes with primary cilia to fluid flow (Supplementary Figure 14f). To investigate the importance of primary cilia in calcium signaling, we cultured cells with EGF, that inhibits the development of cilia and used these cells for the flow experiments (Figure 3b and Supplementary Figure 14j). As shown in Figure 6g and Supplementary Video 8, fluid-flow mediated intracellular calcium signaling is reduced in GCaMP derived non-ciliated cholangiocytes. These data are discussed in the revised manuscript on page 13 line 25. We have also replaced Figure 7a with a better image.

Fifth, the mechanistic part of the rescue of the CFTR function after the treatment with a drug against cystic fibrosis (VX809) combined with an increased fluid flow (page 12) is missing.

We show drug rescue (VX809) with fluid flow in Figure 7h, however flow did not amplify the effect of VX809 as AC1/C2-2. We described these in the manuscript on page 14 line 21.

Sixth, the authors' conclusion on the role of primary cilia in regulation of CFTR in generated cholangiocytes via the cilia-mediated intracellular Ca²⁺ signaling pathway (page 15) is premature and requires additional studies on cholangiocytes in which formation of cilia is prevented.

As indicated above, we have generated cholangiocytes with no cilia and show that intracellular calcium signaling is reduced in these cells (Figure 6g and Supplementary Video 8)

Minor comments:

- In Figure 1a, “Cholangiocyte Day 37” and “Ciliated Cholangiocyte” should be plural – “Cholangiocytes”.

This has been corrected.

- Supplementary Video 1 shows primary cilia in H9(?) cholangiocytes. H9 is confusing, because in the text (page 7) the authors are talking about DHIC5-4D9+. Are DHIC5-4D9+ cholangiocytes H9 cholangiocytes? Needs clarification.

We corrected the figure legend of Supplementary Video 1 explaining these are the human embryonic stem cell (H9 line) derived cholangiocytes. DHIC5-4D9 positivity was also examined in H9 hES derived cholangiocytes. We also add the explanation about this on page 8 line 19 (previously page 7).

- In Figure 3b, the data on “d49 RA” are missing.

If we continue to culture cells with RA until d49, there are no cholangiocytes remaining in the culture plate. Cell viability cannot be sustained in the presence with RA.

- In Figure 5d, e instead of “DMSO” should be “FSK”.

We amended accordingly in Figure 5d and e.

- On page 10 and in Figure 5d, the authors presented data generated on H9 hESC-derived (?) cholangiocyte cysts. What are H9 hESC-derived cholangiocyte cysts? Needs clarification.

H9 is the human embryonic stem cell derived cell line that we used to establish the differentiation protocols.

- The authors need to provide a list of abbreviation that will help the readers to follow the text.

We have added a list of abbreviation.

Reviewer #3 (Remarks to the Author):

In the submitted manuscript the authors describe the production of cholangiocytes from human iPSCs that display an array of functional activities that reproduce key aspects of bile duct function. They go on to show that the cells respond to drugs for the treatment of cystic fibrosis and could form a platform for drug discovery. The study is well conceived and the data are of high quality. Although others have reported the production of bile duct cells from iPSCs previously, this represents a substantial advance in both efficiency and maturity of the cells that are generated. Most importantly the cells produce cilia that have key roles and are known to contribute to cholangiopathies. While enthusiasm for the project is high there are specific areas that could be addressed to further improve the manuscript:

It would be helpful to have a more global profiling of expression to compare to endogenous human cholangiocytes. While the RT-qPCR is convincing, RNA-seq analyses would be an unbiased approach that would provide a greater understanding of the state of the iPSC-cholangiocytes and whether there are changes that could affect function.

We agree with the reviewer and have carried out sc-RNA seq analyses on both the monolayer and 3D cyst-derived populations. The data are shown in Figure 8 and Supplementary Figure 15. We described the results on page 15 of the revised manuscript.

In figure 4 the authors distinguish responses to CFTR drug treatments by iPSC-cholangiocytes from different patients. The patients share the same CFTR variation yet they exhibit distinct responses to the

drug treatments. This is provocative and has the potential to be of value when defining the course of treatments for specific patients. However, there is some concern that the variations do not reflect the patient responses but instead are underlying differences between clonal iPSC lines. It would be useful to examine the response in multiple distinct iPSC clonal cell lines derived from a single individual's (isogenic) somatic cells with a 'normal' CFTR allele. One would predict that such iPSC-derived cholangiocytes have a relatively similar response between clones.

This is an important question that we have addressed by analyzing drug responses on cholangiocytes from 2 independent clones from 2 different CF patient iPSCs (CF01a, CF01b, CF02a and CF02b) carrying the same mutation (total 4 lines). We also differentiated CRISPR-edited mutation corrected lines to cholangiocytes to measure the CFTR response. As we expected, cholangiocytes from corrected lines showed high apical chloride conductance. Responses to VX661/445(s) are missing in CF01a due to the limited access of the small molecules at the time these analyses were done. The results including the clonal differences are shown in Figures 4 and Supplementary Figures 12. We described these findings on page 10 of the revised manuscript.

Minor points:

Some of the labelling of the bar charts was a bit difficult to follow and required a bit of digging to work out what the data represented. Labelling could be clarified throughout the manuscript.

We clarified the labelling in Figures and legends and added the abbreviations in the main text.

In Figure 1B it would be better to use a different color scheme on the bar graphs because red/green can be problematic for those with color-blindness.

We changed the color of bar graphs in Figure 1b.

In Figure 1 C.D., it wasn't intuitive how the authors knew that the CFTR bands were glycosylated with glycosylase treating the extracts. Some clarification could be helpful.

We performed the western blotting again to obtain clearer images and added molecular weight standards besides the figure. Uncropped western bands are also included in the additional files.

In Figure 4, are the microtiter wells referred to by the authors equivalent to the 96-well plates used to determine the z-factor.

Yes, we used 96-well plates for the z-factor determination.

In Figure 5 the swelling response comparing mutant to wild type cells is convincing; however, it wasn't clear whether the relatively subtle changes would be quantitative enough to differentiate between potential treatment options. It could be helpful to show a dose response curve relating swelling to drug treatment.

We agree that the swelling response was subtle in the 3D functional assay. This was one of the reasons we have conducted the apical chloride measurement in the 96 well plates as a direct measurement of CFTR function. With this assay, we could observe difference between the different drugs. As the doses of the drugs used in this paper are widely used in the CF functional assays and did not move forward with the dose response studies.

REVIEWER COMMENTS

Reviewer #1 (Remarks to the Author):

No further comments.

Reviewer #2 (Remarks to the Author):

The authors responded to all my comments and improved the quality of the manuscript.

Reviewer #3 (Remarks to the Author):

The authors have included several new pieces of data that support their conclusions and show that their data represent a substantial advance. I think this study will be received with enthusiasm.

Reviewer #4 (Remarks to the Author):

In the manuscript "Generation of functional ciliated cholangiocytes from human pluripotent stem cells" by Ogawa et al. the authors describe a novel protocol for differentiation of human pluripotent stem cells (hPSC) to cholangiocytes. They claim that this new differentiation strategy overcomes drawbacks of current protocols and reaches a higher level of maturation. This includes high levels of CFTR and the presence of primary cilia of sensing flow thereby providing a model for testing drug efficacy and studying the role of cilia in cholangiocytes.

In the revised manuscript, the authors added single cell RNA sequencing (scRNA-seq) data of the cholangiocytes from the monolayer (2D) and the cyst (3D) cell culture systems using the droplet-based 10x Genomics' Chromium scRNA-seq technology. Since this reviewer was specifically invited to review the scRNA-seq experiment, comments are limited to this.

1. The experimental design includes cholangiocytes from the monolayers (2D) and the cysts (3D). As there is no further comparison group included, such as the original cells (hPSCs), cells previously described by the same group without functional cilia and/or the target cells (such as primary human cholangiocytes, conclusions drawn from this experiment are limited to the comparison of these two cell culture systems.

2. The authors identify 11 different clusters in these groups (Fig. 8a) representing potentially different cell states. Apart from a set of cholangiocyte markers and intrahepatic bile duct markers being expressed by all clusters, only cluster 7 is described in detail and annotated as proliferating cholangiocytes (Fig 8 b). Could the other clusters also be assigned to cell states on the basis of differently expressed markers (Fig. 8 c)? Might these clusters partially represent different states of differentiation/maturation? Might this be linked to the functional data obtained in the two cell culture systems?

3. For further analysis, three different groups were defined (Fig. 8g): (clusters 0+1). These clusters were arbitrarily defined based on the proportion of the two samples used (2D and 3D). However only limited information is provided reasoning this selection. In particular, it should be described why only 5 clusters were used for these groups leaving out the 6 other clusters. Moreover, the groups should be renamed to 2D dominant and 3D dominant in Figures 8g, 8h and S15c to prevent confusion with the samples 2D and 3D in Fig. 8b and f, best with specification of the clusters they are comprised of. Additionally, advantages and limitations of this approach compared to directly comparing the cells of the two samples (2D vs. 3D) should be discussed. Furthermore, the authors should include a table of

the distribution of the two samples across all clusters to increase transparency.

4. Differential gene expression analysis comparing the above mentioned groups (2D dominant, 3D dominant, and overlapping) revealed differences in the expression of certain genes, among them AQP1 with high expression in the 3D dominant group (Fig. S15c). Besides, it is remarkable that AQP1 is almost exclusively expressed in the 3D sample as shown in Fig. 8f. This should also be discussed and linked to the other experiments. Does this lack of AQP1 in the 2D monolayers have functional consequences? Might this also be attributed to incomplete differentiation/maturation?

5. Gene ontology analysis (Fig. 8 h) revealed enrichment of cell junction and ECM terms in 2D dominant and ER related terms in 3D dominant clusters. It should be discussed how this results are connected to the other experimental results obtained comparing those two cell culture systems. Might this indicate differences in maturation/differentiation?

6. In the methods section (line 702ff on page 24) the authors should provide additional information concerning the scRNA-seq to ensure reproducibility. First, more details on sequencing should be provided (sequencing kit and platform). Besides, software versions should be reported consistently (e.g. missing for STAR, R, Seurat, batchelor and clusterProfiler). Additionally, the authors should consider citing STAR (Dobin, A. et al. *Bioinformatics* 29, 15–21 (2013)). Furthermore number of samples per group (n=1) should be stated.

Overall this is a very well done and impressive study. However, the conclusion drawn from the scRNA-seq experiment by the authors, that their protocol promotes the generation of mature, functional cholangiocytes (in lines 554ff, page 19), is not sufficiently supported by the data – mainly due to the lack of an adequate comparison group in the experimental design (such as the original cells (hPSCs), cells previously described by the same group without functional cilia and/or the target cells (such as primary human cholangiocytes). Furthermore, scRNA-seq is an unsuitable method to prove functionality of cells (as stated by the authors in line 556). The authors should revise this statement. Yet, an interesting and novel finding of the scRNA-seq data is demonstrated in Fig 8f, showing that both, 2D monolayer and 3D cyts share identity with chlangiocyte gene sets, more precisely with intrahepatic bile duct gene sets. Nevertheless, the reviewer would suggest to move the single cell figure 8 into supplementary data. The authors included other experiments in their study (including comprehensive in vitro testing and in vivo data), which might indeed be suitable to draw the conclusion that this protocol promotes generation of mature and functional cholangiocytes. However, this has to be judged by the other reviewers as functional tests for differentiated cholangiocytes are beyond the area of expertise of this reviewer.

Please find our point-by-point response to the reviewer's comments. The Reviewer's comments are in *Italic* and the changes made to address the questions/concerns are highlighted in yellow in the revised paper.

REVIEWER COMMENTS

Reviewer #4 (Remarks to the Author):

In the manuscript "Generation of functional ciliated cholangiocytes from human pluripotent stem cells" by Ogawa et al. the authors describe a novel protocol for differentiation of human pluripotent stem cells (hPSC) to cholangiocytes. They claim that this new differentiation strategy overcomes drawbacks of current protocols and reaches a higher level of maturation. This includes high levels of CFTR and the presence of primary cilia of sensing flow thereby providing a model for testing drug efficacy and studying the role of cilia in cholangiocytes.

In the revised manuscript, the authors added single cell RNA sequencing (scRNA-seq) data of the cholangiocytes from the monolayer (2D) and the cyst (3D) cell culture systems using the droplet-based 10x Genomics' Chromium scRNA-seq technology. Since this reviewer was specifically invited to review the scRNA-seq experiment, comments are limited to this.

1. The experimental design includes cholangiocytes from the monolayers (2D) and the cysts (3D). As there is no further comparison group included, such as the original cells (hPSCs), cells previously described by the same group without functional cilia and/or the target cells (such as primary human cholangiocytes, conclusions drawn from this experiment are limited to the comparison of these two cell culture systems.

- *To address this question, we compared our data with a published data set from primary human cholangiocyte (MacParland et al, Nature Communications. 2018 Oct 22;9(1):4383. doi: 10.1038/s41467-018-06318-7.) as we felt this would provide an indication of the state of maturation of the cells generated from the hPSCs. The data are presented in Fig 8f, 8g, 8h, Supplementary Figures 16, and Supplementary Table 5. This comparative analysis demonstrated that the 3D dominant population shows a transcriptomic profile similar to that of the adult cholangiocytes and distinct from the cholangiocytes generated in the monolayer cultures. We described these findings in the revised manuscript on page 16 line 448.*

2. The authors identify 11 different clusters in these groups (Fig. 8a) representing potentially different cell states. Apart from a set of cholangiocyte markers and intrahepatic bile duct markers being expressed by all clusters, only cluster 7 is described in detail and annotated as proliferating cholangiocytes (Fig 8 b). Could the other clusters also be assigned to cell states on the basis of differently expressed markers (Fig. 8 c)? Might these clusters partially represent different states of differentiation/maturation? Might this be linked to the functional data obtained in the two cell culture systems?

- *In the revised manuscript, we show the top five genes differentially expressed in all clusters in Supplementary Figure 15a. The list of differentially expressed genes is presented in Supplementary Table 1 (Supplementary Dataset 1 Excel sheet). We have also added a Supplementary Table 2 showing the cell status of all 12 clusters (0-11). Cluster 8, 9, 10 and 11 represent a small subset of cells that uniquely express secretory protein and hormonal genes including *SPARC* (cluster 8), *AGR2* (cluster 9), *CHGA* (cluster10), *GHRL* (cluster 10) and *AZGP1* (cluster 11). As these are genes expressed by gastrointestinal tract secretory cells, these clusters may represent contaminants of these cell types. Cluster 7, as indicated in main text, contains predominantly cycling cells. Cluster 5, the most prominent in the 3D cysts contains cells*

that express genes indicative of hepatocyte/hepatic progenitor cells. These cluster also contained cells that expressed *TFF1*, *TFF2*, *FOS*, *AGR2*, *LGALS4* (Supplementary Figure 15a), and *ACE2* (Figure 8e) indicative of biliary secretory cells. Based on our new analysis, cluster 2 and 3 present in the 3D dominant cluster show expression profiles comparable to adult human cholangiocytes. Taken together, these findings suggest that the major clusters represent cells at different stages of cholangiocyte differentiation whereas some of the smaller clusters contain contaminating gastrointestinal cells. We annotated clusters in Figure 8 accordingly.

3. For further analysis, three different groups were defined (Fig. 8g): (clusters 0+1). These clusters were arbitrarily defined based on the proportion of the two samples used (2D and 3D). However only limited information is provided reasoning this selection. In particular, it should be described why only 5 clusters were used for these groups leaving out the 6 other clusters. Moreover, the groups should be renamed to 2D dominant and 3D dominant in Figures 8g, 8h and S15c to prevent confusion with the samples 2D and 3D in Fig. 8b and f, best with specification of the clusters they are comprised of. Additionally, advantages and limitations of this approach compared to directly comparing the cells of the two samples (2D vs. 3D) should be discussed. Furthermore, the authors should include a table of the distribution of the two samples across all clusters to increase transparency.

- We have included a table (Supplementary Table 3) in the revised manuscript showing the distribution of the two samples across all clusters. Additionally, populations in the previous figures were renamed as 2D dominant and 3D dominant to prevent confusion. Cluster 5 was excluded due to the expression of progenitor marker AFP and cluster 7 was excluded due to the large proportion of proliferative cells. Clusters 8 to 11 were not included in the analysis as they contained small number of gastrointestinal secretory cells. We selected the **2D dominant** population made up of clusters 4 and 6, the **3D dominant** population consisting of clusters 2 and 3, and the **combined** population represented by clusters 0 and 1 for further investigation. We described our reasons for selecting these clusters in the revised manuscript on page 15 line 442. The differentially expressed genes among these 3 populations are listed in Supplementary Table 4 (Supplementary Dataset 1 Excel sheet). Additionally, we compared the transcriptional profile of these 3 populations to the profile of cholangiocytes isolated from the human adult liver (Supplementary Table 5).

4. Differential gene expression analysis comparing the above mentioned groups (2D dominant, 3D dominant, and overlapping) revealed differences in the expression of certain genes, among them AQP1 with high expression in the 3D dominant group (Fig. S15c). Besides, it is remarkable that AQP1 is almost exclusively expressed in the 3D sample as shown in Fig. 8f. This should also be discussed and linked to the other experiments. Does this lack of AQP1 in the 2D monolayers have functional consequences? Might this also be attributed to incomplete differentiation/maturation?

- Lower expression of the *AQP1* in monolayer (Cluster 4 and 6 in Figure 8a) may be indicative of a less mature population. This interpretation is supported by the comparison of our samples (monolayer and cysts) to primary adult cholangiocyte that showed a higher degree of similarity between 3D dominant population and the adult cells than between the 2D monolayer population and the adult cells. *AQP1* is water selective channel protein (Marinelli and LaRusso et al.; <https://doi.org/10.1074/jbc.272.20.12984>), that is predominantly expressed in cholangiocytes and sinusoidal endothelial cells in the liver. The higher levels of expression in the 3D cysts than in the monolayer may indicate that some aspect of cyst formation including stimulation through orbital shaking or the formation of cyst structure may play a role in the upregulation of its expression. Further studies would be required to determine why *AQP1* is expressed at higher levels in the 3D cysts than in the monolayers.

5. *Gene ontology analysis (Fig. 8 h) revealed enrichment of cell junction and ECM terms in 2D dominant and ER related terms in 3D dominant clusters. It should be discussed how this results are connected to the other experimental results obtained comparing those two cell culture systems. Might this indicate differences in maturation/differentiation?*

- These differences are of interest and may reflect difference in the state of maturation or may reflect differences in culture format, 2D vs 3D used to generate the cells. The differences indicate that 2D format is likely more supportive of ECM deposition that is involved in cell proliferation, adhesion, migration and cell differentiation (Lu et al, doi: 10.1101/cshperspect.a005058). By contrast, 3D format supports the development of cells that express ER related gene associated with protein synthesis, lipid biogenesis and calcium regulation (Schwarz and Blower et al, doi: 10.1007/s00018-015-2052-6). We have added comments on these differences in the Discussion on page 20 line 572.

6. *In the methods section (line 702ff on page 24) the authors should provide additional information concerning the scRNA-seq to ensure reproducibility. First, more details on sequencing should be provided (sequencing kit and platform). Besides, software versions should be reported consistently (e.g. missing for STAR, R, Seurat, batchelor and clusterProfiler). Additionally, the authors should consider citing STAR (Dobin, A. et al. Bioinformatics 29, 15–21 (2013)). Furthermore number of samples per group (n=1) should be stated.*

- We included all detailed information in the method section and reporting summary. The suggested paper was cited in the reference and the number of samples is indicated in the method section.

Overall this is a very well done and impressive study. However, the conclusion drawn from the scRNA-seq experiment by the authors, that their protocol promotes the generation of mature, functional cholangiocytes (in lines 554ff, page 19), is not sufficiently supported by the data – mainly due to the lack of an adequate comparison group in the experimental design (such as the original cells (hPSCs), cells previously described by the same group without functional cilia and/or the target cells (such as primary human cholangiocytes). Furthermore, scRNA-seq is an unsuitable method to prove functionality of cells (as stated by the authors in line 556). The authors should revise this statement. Yet, an interesting and novel finding of the scRNA-seq data is demonstrated in Fig 8f, showing that both, 2D monolayer and 3D cyts share identity with chlangiocyte gene sets, more precisely with intrahepatic bile duct gene sets. Nevertheless, the reviewer would suggest to move the single cell figure 8 into supplementary data. The authors included other experiments in their study (including comprehensive in vitro testing and in vivo data), which might indeed be suitable to draw the conclusion that this protocol promotes generation of mature and functional cholangiocytes. However, this has to be judged by the other reviewers as functional tests for differentiated cholangiocytes are beyond the area of expertise of this reviewer.

- We agreed that scRNA seq is not an appropriate method to demonstrate the functionality. However, the comparative analyses showing similar molecular profiles in the 3D hPSC-derived population and the adult cholangiocytes strongly suggests that the cells generated in vitro are mature. We feel that these findings are novel and important and therefore have chosen to show them in the main Figure 8. We have modified the Discussion and eliminated the reference to functionality based on the scRNA seq results. The function of hPSC-derived cholangiocytes was confirmed in the other sections of this paper. The other 3 reviewers felt that the approaches we used demonstrate that the hPSC-derived cells displayed appropriate cholangiocyte function.

REVIEWERS' COMMENTS

Reviewer #4 (Remarks to the Author):

I thank the authors for their through response and revisions to the manuscript. The authors have addressed the points I have raised, I am happy to support publication.